# Nanoparticle-based DNA vaccine protects against SARS-CoV-2 variants in female preclinical models

A safe and effective vaccine with long-term protection against SARS-CoV-2 variants of concern (VOCs) is a global health priority. Here, we develop lipid nanoparticles (LNPs) to provide safe and effective delivery of plasmid DNA (pDNA) and show protection against VOCs in female small animal models. Using a library of LNPs encapsulating unique barcoded DNA (b-DNA), we screen for b-DNA delivery after intramuscular administration. The top-performing LNPs are further tested for their capacity of pDNA uptake in antigen-presenting cells in vitro. The lead LNP is used to encapsulate pDNA encoding the HexaPro version of SARS-CoV-2 spike (LNP-HPS) and immuno-genicity and protection is tested in vivo. LNP-HPS elicit a robust protective effect against SARS-CoV-2 Gamma (P.1), correlating with reduced lethality, decreased viral load in the lungs and reduced lung damage. LNP-HPS induce potent humoral and T cell responses against P.1, and generate high levels of neutralizing antibodies against P.1 and Omicron (B.1.1.529). Our findings indi-cate that the protective efficacy and immunogenicity elicited by LNP-HPS are comparable to those achieved by the approved COVID-19 vaccine from Bion-tech/Pfizer in animal models. Together, these findings suggest that LNP-HPS hold great promise as a vaccine candidate against VOCs.

A safe and effective vaccine with long-term protection against severe acute respiratory syndrome coronavirus 2 (SARS-CoV-2) and variants of concern (VOCs) is an urgent global health priority[1]. More than 641 million cases of coronavirus disease 2019 (COVID-19) caused by SARS-CoV-2 have been confirmed since its emergence in December 2019, with more than 6.6 million deaths worldwide (as of February 2023)[2]. To date, several COVID-19 vaccines based on different technologies are currently approved[3]. Among these, the two most widely used world-wide are based on messenger RNA (mRNA) encapsulated in lipid nanoparticles (LNPs): mRNA-1273[4] and BNT162b2[5] from Moderna and Pfizer-BioNTech, respectively[6]. When compared to conventional vac-cines, lipid-based nanotechnology demonstrated remarkable efficacy (above 94%), establishing the potential of this approach to combat the existing SARS-CoV-2 and its variants, including those yet to come[6,7].

Although nanotechnology-based vaccines have been reported to trigger both cellular and humoral immune responses[8–10], emerging SARS-CoV-2 VOCs increase the demand for novel strategies. In this regard, many researchers and companies have been working on next-generation vaccines using new technologies and platforms, including updated versions of approved vaccines to protect against broader coronavirus classes and ensure long-lasting immunity[11].

Beyond mRNA, nanoparticle-based DNA vaccines can also be leveraged to train the immune system to selectively attack viruses, with a coordinated response of CD4[+] and CD8[+] T cells and broadly neutralizing antibodies (nAbs)[12,13]. In addition, DNA is cost-effective, easy to produce, and more stable than mRNA, which can be relevant for remote areas and developing nations[12,14]. While ZyCoV-D is the sole DNA vaccine approved for COVID-19[15,16], there are at least 10 others currently in various stages of clinical trials (Phase I-III), including can-didates like INO-4800, AG0301-COVID-19, and GX-19N[17,18]. This diver-sity of candidates demonstrates the maturity of DNA vaccine technology.

✉e-mail: ppiresgo@reitoria.ufmg.br

We previously reported results for RNA[19–21] and DNA[22,23] delivery platforms using LNPs. Here, we developed a scalable nanoparticle-based DNA vaccine against SARS-CoV-2 VOCs. Our goal was to design an LNP platform for the safe and effective delivery of DNA to enhance antigen presentation and immune reactivity against SARS-CoV-2 variants. To achieve this aim, we designed a library of 15 engineered LNPs that encapsulate custom-designed barcoded DNA (b-DNA). We screened these multiple b-DNA LNPs in different tissues and cells of interest to select a lead LNP to induce potent immune responses against COVID-19 (Fig. 1A). Next, we assessed the protective efficacy of our leading LNP, namely LNP-HPS, formulated with recombinant HexaPro spike plasmid DNA against SARS-CoV-2 Gamma lineage (P.1). Additionally, we evaluated the immunogenicity against SARS-CoV-2 VOCs P.1 and Omicron (B.1.1.529) (Fig. 1B). Our findings reveal that a two-dose immunization of our lead LNP-based DNA vaccine elicited a robust protective effect against P.1 and induced production of IFN-γ and granzyme-B response by T cells in K18-hACE2 mice. Importantly, it also led to a reduced viral load in the lungs and enhanced antigen-specific IgG titers in both mice and hamsters. Furthermore, LNP-HPS generated high levels of neutralizing antibodies against SARS-CoV-2 VOCs P.1 and Omicron in both animal models. Together, our data demonstrate the versatility of our nanoparticle-based DNA vaccine platform, suggesting its potential utility in preventing SARS-CoV-2 infection.

## Results

### Preparation and characterization of b-DNA-LNPs

To investigate the LNP formulation excipients that enable DNA delivery across multiple organs of interest, such as spleen, we developed a library of 15 engineered LNPs formulated by combining the ionizable lipid, distinct helper lipids (DOPE, DOTAP, or DSPC), cholesterol, and lipid-anchored poly (ethylene glycol) at varying molar ratios (Supplementary Table 1) with unique b-DNAs via controlled microfluidic mixing (Fig. 1A). A library of 15 b-DNA-LNPs was formulated by (i) varying the helper lipids and their molar percentage in each formulation and (ii) varying molar percentage of cholesterol in each formulation. Each LNP was made to encapsulate a unique b-DNA to be detected in each organ via deep sequencing (Supplementary Table 2). Dynamic light scattering (DLS) measurements varied according to the type of auxiliary lipid used: for DOPE, the hydrodynamic diameter ranged from 106-127 nm; DSPC, 96-233 nm; DOTAP, 84-162 nm (Supplementary Table 3). The zeta potential values of the LNPs containing DOPE varied between −1.41 mV and −9.37 mV, DSPC between −12.35 mV and 1.46 mV, and DOTAP between −7.84 mV and 27.53 mV (Supplementary Table 3). Following characterization, the library of 15 unique LNPs was used to identify top-performing LNPs for DNA delivery in lymphoid tissues, such as spleen, as described below.

### b-DNA-LNPs to target lymphoid tissues

The 15 LNP formulations encapsulating unique DNA barcodes (b-DNA-LNPs) were pooled together and administered via intramuscular injection into mice. Tissues including the liver, spleen, lungs, heart, muscle and lymph node were harvested 4 hours post-injection and DNA was extracted from these tissues, with LNP delivery to multiple organs accurately quantified through deep sequencing (Fig. 1). The barcodes from each tissue were amplified by PCR and analyzed by deep sequencing to assess the relative biodistribution of each b-DNA-LNP formulation in these organs (Fig. 1C–E and Supplementary Fig. 1). The normalized delivery quantification reflects how efficiently each barcode was delivered to each specific organ, relative to all other injected barcoded LNPs in the same organ.

After evaluating the biodistribution of 15 formulations, we then selected two top-performing LNPs to proceed with in vitro transfection assays with different cell types (Fig. 2A). Based on the delivery results, B2, B3, B4, and B10 LNPs demonstrated enhanced DNA delivery to the spleen, muscle and lymph node (Fig. 1C, D and Supplementary Fig. 1A, B).

Because in our approach we are looking for improved specificity, B2 and B3 LNP were not selected due to its concomitant enhanced delivery to the liver and heart, respectively (Supplementary Fig. 1C, D). Thus, B4 and B10 LNPs were selected as the top-performing LNPs in light of specificity to the spleen and lymph nodes and minimized DNA delivery to the heart and liver compared to other screened LNPs[24,25].

### In vitro transfection efficiency of lead LNPs

To validate the b-DNA-LNP screening platform, we investigated the transfection efficiency of top-performing LNP formulations in antigen-presenting cells (macrophages, dendritic cells, and B cells), myoblasts, fibroblasts, and endothelial cells (Fig. 2, and Supplementary Fig. 2). To determine the optimal concentration with the highest transfection efficiency and lowest toxicity in different cell types, we formulated LNP B4 with pDNA encoding ZsGreen. All the cell types of interest for the vaccine were treated with LNP B4 at varying concentrations of pDNA from 0.00625-0.8 μg/well for evaluation of GFP expression and cell viability at 24 h, 48 h and 72 h (Fig. 2B–D, F–H, J–L, N–P and Supplementary Fig. 2B–D). We found that 0.4 μg/well and 0.2 μg/well of DNA loaded in LNP B4 were the lower concentrations which resulted in a higher transfection efficiency without cytotoxicity when compared to the other groups in antigen-presenting cells (Fig. 2B–I). For fibroblasts, myoblasts and endothelial cells these optimal concentrations were 0.2/well μg, 0.05 μg/well and 0.1 μg/well, respectively (Fig. 2J–Q and Supplementary Fig. 2). Subsequently, we compared the top-performing lead LNP formulations B4 and B10 in each cell line at the optimal concentrations for each cell. Although LNPs B4 and B10 successfully transfected the different cells types of interest, LNP B4 exhibited significantly enhanced fluorescence compared to LNP B10 (Fig. 2E, I, M Q, and Supplementary Fig. 2E–G). Because our goal was to design a LNP platform for safe and effective pDNA delivery to antigen-presenting cells, we selected LNP B4 for further experiments.

### Synthesis and characterization of LNP-based DNA vaccine

After identifying the top-performing LNP formulations for DNA delivery in spleen and enhanced transfection in antigen-presenting cells, we next formulated our lead LNP encapsulating recombinant HexaPro Spike plasmid DNA (LNP-HPS) or control plasmid DNA (LNP-C) (Fig. 3A and Supplementary Fig. 3) to assess efficacy and immunogenicity against SARS-CoV-2 variants of Gamma (P.1) and Omicron (B.1.1.529) lineage. Cryo-TEM revealed that the LNPs were monodisperse with a solid structure (electron-dense nucleus evidencing compact interiors) (Fig. 3B, C). DLS measurements showed homogenous size distribution for each LNP formulation with hydrodynamic diameters of 146 ± 6 nm and 161 ± 12 nm, polydispersity index of 0.149 ± 0.035 and 0.177 ± 0.02 (Fig. 3D), and slightly negative zeta potentials of −5.09 ± 2.12 and −5.98 ± 2.41 (Fig. 3E) for LNP-C and LNP-HPS, respectively. We also measured the pKa of LNP-C and LNP-HPS formulations via TNS assay, which demonstrated pKa values of 7.06 and 7.43, respectively (Supplementary Fig. 4). Further, we also evaluated the protective effect of LNP against DNA degradation via DNase activity assay (Fig. 3F). Naked pDNAs without DNase were used as a positive control. Free pDNA-C and pDNA-HPS + DNase I were susceptible to enzymatic degradation, evidenced by the absence of a bright DNA band. pDNA was not observed when loaded in LNP formulations. After the disruption of LNPs using isopropanol, pDNA was exposed. Disrupted LNPs without DNase I were used as a positive control. Compared to controls, pDNAs encapsulated in LNPs (LNP-C and LNP-HPS) and incubated with DNase I were adequately protected from degradation by LNPs (Fig. 3F). We also demonstrated the ability of LNP-HPS to induce enhanced Spike protein expression in HEK-293 cells (Fig. 3G–H).

### Vaccine efficacy in K18-hACE2 mice

To determine the efficacy of LNP-based DNA vaccine, K18-hACE2 mice were intramuscularly vaccinated with two doses of PBS, LNP-C, naked

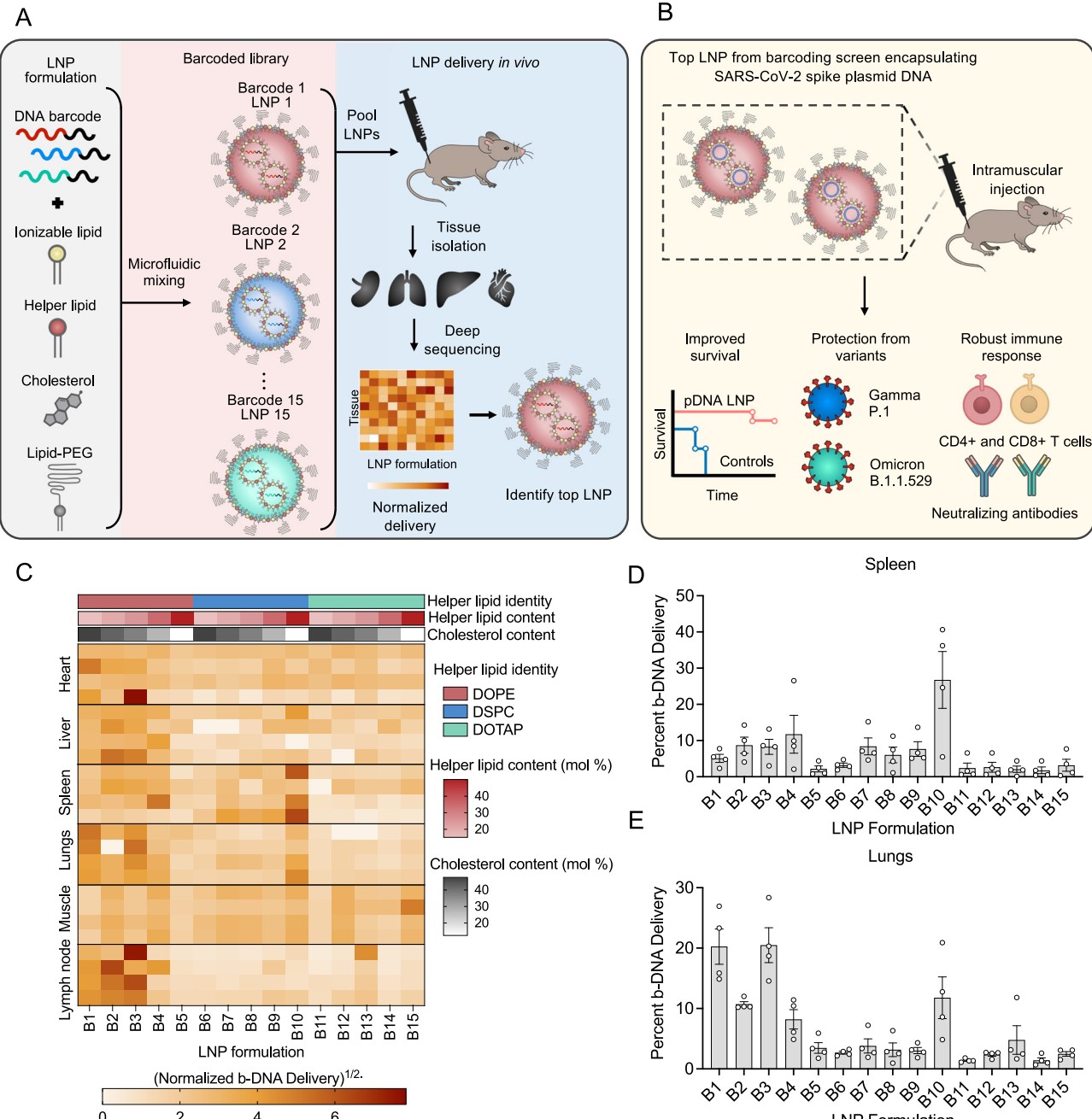

**Fig. 1 | Lipid nanoparticles (LNPs) encapsulating barcoded pDNA (b-DNA) for accelerated in vivo delivery screening and identification of lead formulations.**
**A** Schematic of LNPs encapsulating barcoded DNA (b-DNA) for accelerated in vivo delivery screening. Left: LNPs were formulated via microfluidic mixing of an aqueous phase of b-DNA and an ethanol phase of lipids. Middle: Schematic formation of LNPs encapsulating b-DNA. LNPs were formulated via microfluidic mixing, and each LNP formulation was encapsulated with a unique b-DNA. Right: 15 LNP formulations (B1–B15) were formulated by varying the identity and molar ratio of phospholipid (DOPE, DOTAP, or DSPC) and molar ratio of cholesterol. A 0.5 μg b-DNA dose of each b-DNA-LNP was pooled and administered to C57BL/6 mice intramuscularly as a single injection. 4 h post injection, b-DNA delivery to the heart, liver, spleen, lung, draining lymph nodes, and muscle were quantified. Organs were harvested 4 h post-injection, and b-DNA delivery was quantified using both whole-organ for deep sequencing to identify the top-performing LNP. **B** Schematic of

immunization with LNP-based DNA vaccine and the efficacy in K18-hACE2 mice against SARS-CoV-2 variants. Top: K18-hACE2 mice were immunized with either controls or LNP-HPS and received a booster dose after 3 weeks. Bottom: vaccinated mice displayed decreased lethality post-viral challenge and generated potent cellular and humoral responses against SARS-CoV-2 VOCs, indicating a strong and comprehensive immune response following vaccination. **C** Darker clusters indicate higher delivery, whereas lighter clusters indicate lower delivery. Within the heatmap, the delivery of different LNP formulations within the same organ (left to right) can be compared. The delivery of the same LNP formulation across different samples (top to bottom) cannot be compared. **D**, **E** The bar graph illustrates the percentage quantification of LNP (B1–B15) delivery in different target tissues 4 hours after intramuscular injection. Data are presented as mean ± SEM ($n = 4$/ group). Source data are provided as a Source Data file.

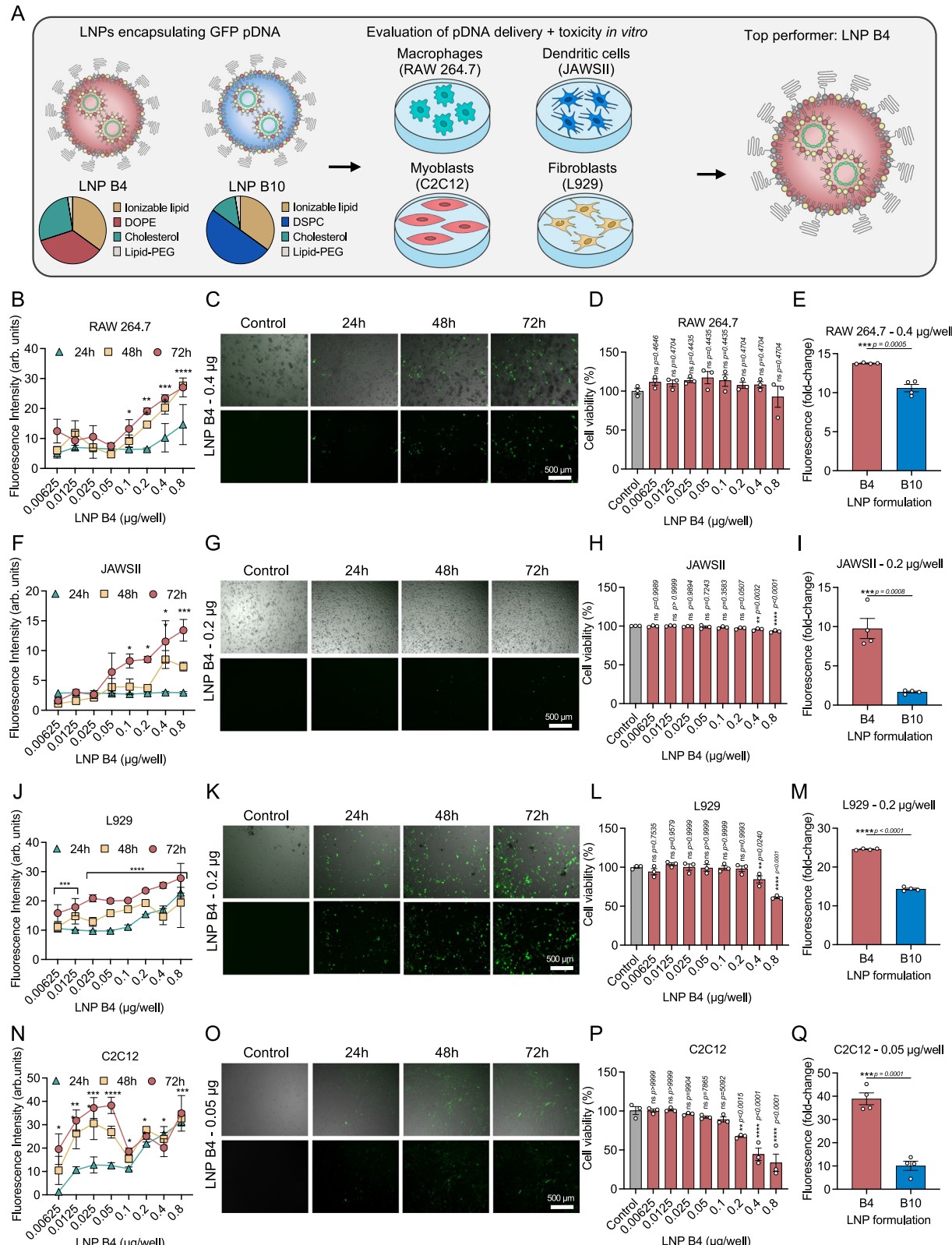

HexaPro Spike plasmid DNA (pHPS), LNP-HPS or approved COVID-19 vaccine from Biontech/Pfizer (BNT-mRNA) at 21 days apart (Fig. 4A). Mice were given a lethal challenge of $6 \times 10^4$ PFUs of SARS-CoV-2 variant Gamma (P.1) 15 days after the second dose of immunization. Notably, the survival rates were 85% and 100% for mice immunized with LNP-HPS and BNT-mRNA, respectively, in contrast to the LNP-C, pHPS, and PBS control groups, which exhibited 100% lethality within

7 days post-challenge (Fig. 4B). Also, mice vaccinated with LNP-HPS showed significantly less weight loss at days 3–6 after challenge compared to LNP-C, pHPS, and PBS control groups, and also recovered their pre-challenge weight at day 6 (Fig. 4C). Importantly, there was no significant difference in lethality and weight loss between BNT-mRNA, LNP-HPS, and Mock. Therefore, weight measurements demonstrated a correlation with lethality, as mice vaccinated with LNP-HPS and BNT-

**Fig. 2 | Top-performing LNPs induced enhanced in vitro GFP expression in different cell lines. A** Schematic of the 2 top-performing LNPs encapsulating DNA encoding GFP for in vitro screening of transfection efficiency in four different cell lines. **B, F, J, N** Quantification of GFP expression was measured after 24 h, 48 h and 72 h, and cell viability was measured after 72 h in **D** macrophages (RAW 264.7), **H** dendritic cells (JAWSII), and **L** fibroblasts (L929), and **P** myoblasts (C2C12) transfected with LNP B4 at different doses (n = 3/group). **C, G, K, O** Representative GFP fluorescence (Bottom) and overlaid on brightfield (Top) photomicrographs after treatment of LNP B4 in four different cell lines (n = 4/group). **E, I, M, Q** Quantification of GFP expression after treatment with B4 and B10 LNPs in different cell lines. Data are presented as mean ± SEM (n = 4/group). Data are presented as mean ± SEM. **B, F, J, N, D, H, L, P** One-way ANOVA with Dunnet's post hoc test compared to control; ns not significant, *p < 0.05, **p < 0.01, ***p < 0.001 ****p < 0.0001. **E, I, M, Q** Two-tailed unpaired t-test; ns not significant, **p < 0.01, ***p < 0.001, ***p < 0.0001. Source data are provided as a Source Data file.

mRNA were protected, whereas the LNP-C, pHPS and PBS control groups revealed weight loss starting from 3 days post infection. To further investigate the viral load in the lungs, all immunized groups were euthanized at day 5 after challenge with our lethal model using $6 \times 10^4$ SARS-CoV-2 variant Gamma lineage (P.1) (Fig. 4D). As expected, the viral load was significantly lower in mice vaccinated with LNP-HPS and BNT-mRNA, in contrast to those in the LNP-C, pHPS, and PBS control groups as detected by plaque-forming (Fig. 4E, F) and RT-qPCR (Fig. 4G). We also used an immunofluorescence assay to assess the presence of spike (S) glycoprotein in the lungs of immunized mice. Reinforcing previous viral load results, the presence of spike (S) glycoprotein was significantly reduced in mice vaccinated with LNP-HPS and BNT-mRNA compared to LNP-C, pHPS, and PBS control groups (Fig. 4H, I). In addition, we observed that the reduced viral load in the lungs of mice immunized with LNP-HPS were comparable to BNT-mRNA. Together, our results indicated a partial, but significant protection of our LNP-based DNA vaccine against SARS-CoV-2 variant Gamma lineage (P.1), with a level of protection comparable to that achieved by BNT-mRNA.

### Histopathological analysis of immunized K18-hACE2 mice
To assess the lung pathology damage and inflammatory score, we performed histopathological analysis in lungs of control and immunized mice at 5 dpi (Fig. 5A). In infected mice given PBS, pHPS or vaccinated with LNP-C, there was intense and diffuse infiltrate in the pleura and subpleural spaces, in the bronchi and alveoli (Fig. 5B–F). There was also edema and thickened alveolar walls and perivascular infiltrates. In mice immunized with LNP-HPS or BNT-mRNA, there was an overall preservation of pulmonary architecture. Indeed, there was a much less diffuse interstitial pneumonia, characterized by moderate inflammatory infiltrate around the bronchi and discrete infiltrate around the vessels and in the alveolus for both LNP-HPS or BNT-mRNA groups (Fig. 5B–F). Importantly, no damage or histological changes were observed in other tissues such as brain, kidney, heart and liver for all groups compared to Mock (Supplementary Fig. 7B).

### Humoral response in K18-hACE2 mice
To assess the humoral response induced by LNP-based DNA immunization, we measured antigen-specific IgG titers and levels of neutralizing antibodies pre and post infection in sera and IgA titers post infection in lung tissue against SARS-CoV-2 variants P.1 and Omicron (Fig. 6). For this, all immunized groups were euthanized 5 days post infection (Fig. 6A). We found significantly enhanced both anti-S IgG titers in serially diluted pre-infection sera (1:400-1:12800) (Fig. 6B) and post infection sera (1:400–1:6400) (Fig. 6C) as well as increased anti-S IgA titers (Fig. 6D) in the lungs of mice post infection immunized with LNP-HPS or BNT-mRNA compared to LNP-C, pHPS and PBS control groups. We next evaluated the levels of neutralizing antibodies in pre and post infection sera of immunized mice against SARS-CoV-2 variants P.1 and Omicron. The neutralization response elicited by immunization with LNP-HPS prior to infection against P.1 (PRNT-50% titer = 1:620) (Fig. 6E, F) and Omicron (PRNT-50% titer = 1:620) (Fig. 6I, J) was significantly higher than that observed in the LNP-C, pHPS, and PBS control groups. Importantly, no significant difference was found between the neutralization induced by

LNP-HPS and BNT-mRNA group. Similar results were found for post infection sera neutralizing titers against P.1 (PRNT-50% titer = 1:620) (Fig. 6G, H) and Omicron (PRNT-50% titer = 1:1240) (Fig. 6K, L) with both LNP-HPS and BNT-mRNA groups exhibiting higher P.1 and Omicron neutralization compared to LNP-C, pHPS, and PBS control groups. Together, the findings substantiate the immunogenicity of LNP-HPS and reveal a humoral response that is comparable to that elicited by BNT-mRNA vaccination.

### T cell response
To investigate T cell response in vaccinated mice, spleens from immunized mice were harvested 5 days post infection and stimulated with 10 µg/mL of RBD-S1 Spike protein (Fig. 7A). Flow cytometric analyses were performed to characterize cell infiltrates (Fig. 7B, E) and cytokine production from CD4+ and CD8+ T cell subpopulations, along with memory subsets (Fig. 7, and Supplementary Fig. 5A). There was enhanced production of IFN-ɣ and Granzyme-B (Gran-B) in CD4+ and CD8+ T cells of mice vaccinated with LNP-HPS or BNT-mRNA, as compared to LNP-C, pHPS, and PBS control groups (Fig. 7C, D, F, G). Furthermore, we observed an increase of central and effector/effector memory subsets in CD8+ T cells (Fig. 7H, N), and increased effector/effector memory CD4+ T cells subpopulations (Fig. 7Q) triggered by the immunization with LNP-HPS or BNT-mRNA. Importantly, there was also an upregulation of IFN-ɣ in central memory subsets in CD8+ (Fig. 7I) and CD4+ (Fig. 7L) T cells subpopulations as well as upregulation of IFN-ɣ and Gran-B in effector/effector memory CD4+ T cells subsets (Fig. 7R, S, respectively) induced by immunization with LNP-HPS or BNT-mRNA. Furthermore, an enhanced Gran-B in central subsets of CD8+ and CD4+ T cells was found in mice immunized with LNP-HPS (Fig. 7J, M). Therefore, the T cell response elicited by immunization with LNP-HPS is comparable to the data of BNT-mRNA.

### Levels of cytokines and chemokines
To assess the levels of cytokines and chemokines in immunized mice, sera and lung tissue were collected five days post infection. Using flow cytometry, we examined the serum levels of IL-6, IL-10, CCL2, IFN-γ, TNF, and IL-12p70 (Supplementary Fig. 5). Notably, the LNP-HPS, BNT-mRNA, and pHPS immunized groups exhibited decreased levels of IL-6 and IL-12p70 compared to the PBS, LNP-C groups (Supplementary Fig. 5B, G). However, no differences were observed in the levels of IL-10, CCL2, IFN-γ, and TNF between groups (Supplementary Fig. 5C–F). Additionally, mRNA expression of cytokines and chemokines in lung tissue was assessed using RT-qPCR (Supplementary Fig. 6). No changes in mRNA levels of IL-10 and TNF were observed between LNP-HPS, PBS, LNP-C, and pHPS. In contrast, the BNT-mRNA group displayed enhanced mRNA levels of IL-10 and TNF compared to all other groups (Supplementary Fig. 6D, H). Increased mRNA levels of CXCL14 were also found for LNP-HPS, BNT-mRNA, pHPS, and LNP-C compared to the PBS group (Supplementary Fig. 6E). No differences were noted in the mRNA levels of IL-1β, IL-17, IL-6, and IFN-γ between all groups (Supplementary Fig. 6B, C, F, G, and I). Collectively, no correlation was identified between the levels of cytokines and chemokines and the immunogenicity induced by LNP-HPS vaccination.

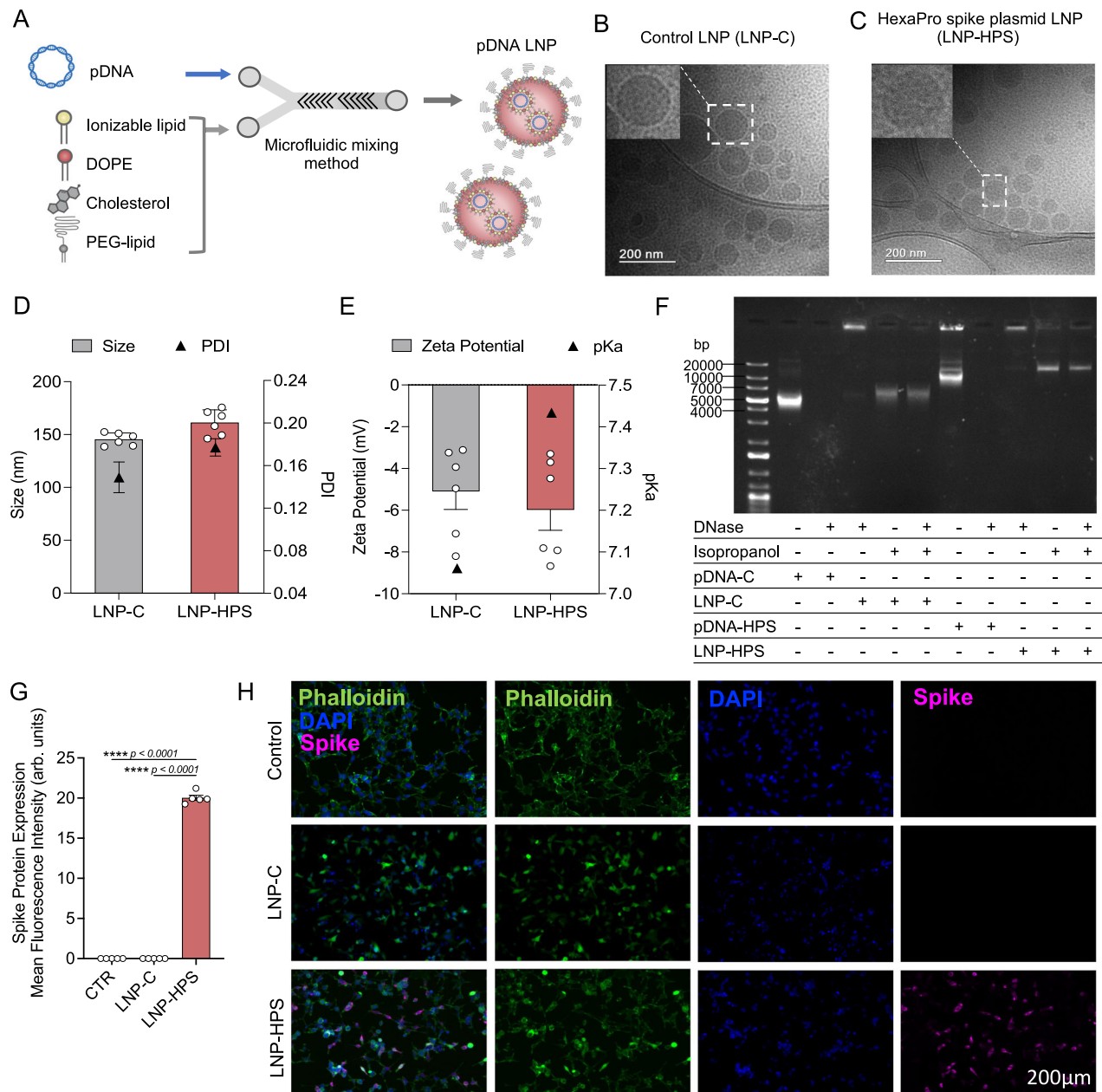

**Fig. 3 | Schematic of rational design and characterization of LNP-based DNA vaccine formulation. A** Traditional four-component LNPs, which consist of ionizable lipid component combined with a phospholipid to support bilayer structure, cholesterol to enhance bilayer stability, and a lipid-anchored-PEG to increase circulation time prepared via microfluidic mixing method. Representative cryo-TEM microscopy of LNPs encapsulated with pDNA: **B** LNP-C **C** LNP-HPS (scale bar, 200 nm). **D** Average hydrodynamic diameter and polydispersity index (PDI) of LNPs, measured by dynamic light scattering (DLS) ($n = 6$/group). **E** Zeta potential showed slightly negatively charged LNPs. pKa values of these LNPs varied from 7 to 7.4 ($n = 6$/group). **F** Agarose gels of LNPs after DNase activity assay reveals preserved pDNA when encapsulated in LNPs. **G, H** Spike protein expression in HEK-293 cells treated with LNP-C and LNP-HPS after 48 h via immunofluorescence ($n = 5$/group). Representative fluorescence images of HEK-293 cells marked with anti-Spike antibody. Samples were stained with anti-SARS-CoV-2 Spike protein (RBD) antibody (magenta), Phalloidin (green), and DAPI (blue), acquired using a ×10 objective. Data are presented as mean ± SEM; ****$p < 0.0001$. **G** One-way ANOVA followed by Tukey's multiple comparison test. **B**–**H** Each experiment was repeated at least three times independently with similar results and the representative dataset is presented. Source data are provided as a Source Data file.

## Immunogenicity in Syrian Hamsters

To assess and validate the immunogenicity of the LNP-HPS in a larger animal model, Syrian hamsters were intramuscularly vaccinated with two doses of of PBS, LNP-C, pHPS, LNP-HPS or BNT-mRNA at 21 days apart. Hamsters were given a challenge of $6 \times 10^5$ PFUs of SARS-CoV-2 variant Gamma (P.1) at 15 days after second dose of immunization and euthanized at day 5 after challenge (Fig. 8A). Hamsters vaccinated with LNP-HPS or BNT-mRNA exhibited no weight loss from days 3 to 5 post-challenge, in contrast to the LNP-C, pHPS, and PBS control groups.

(Fig. 8B). As expected, hamsters vaccinated with LNP-HPS or BNT-mRNA displayed a significantly lower viral load compared to those in the LNP-C, pHPS, and PBS control groups, as evidenced by RT-qPCR (Fig. 8C) and plaque-forming assays (Fig. 8D, E). To assess the lung damage and inflammatory score, we performed histopathological analysis of hamster lungs at 5 days post infection (Fig. 8F). Infected hamsters immunized with PBS, pHPS, or LNP-C exhibited a pronounced and widespread infiltrate in the pleura, subpleural spaces, bronchi, and alveoli bronchi, and alveoli (Fig. 8G–J). Additionally, there

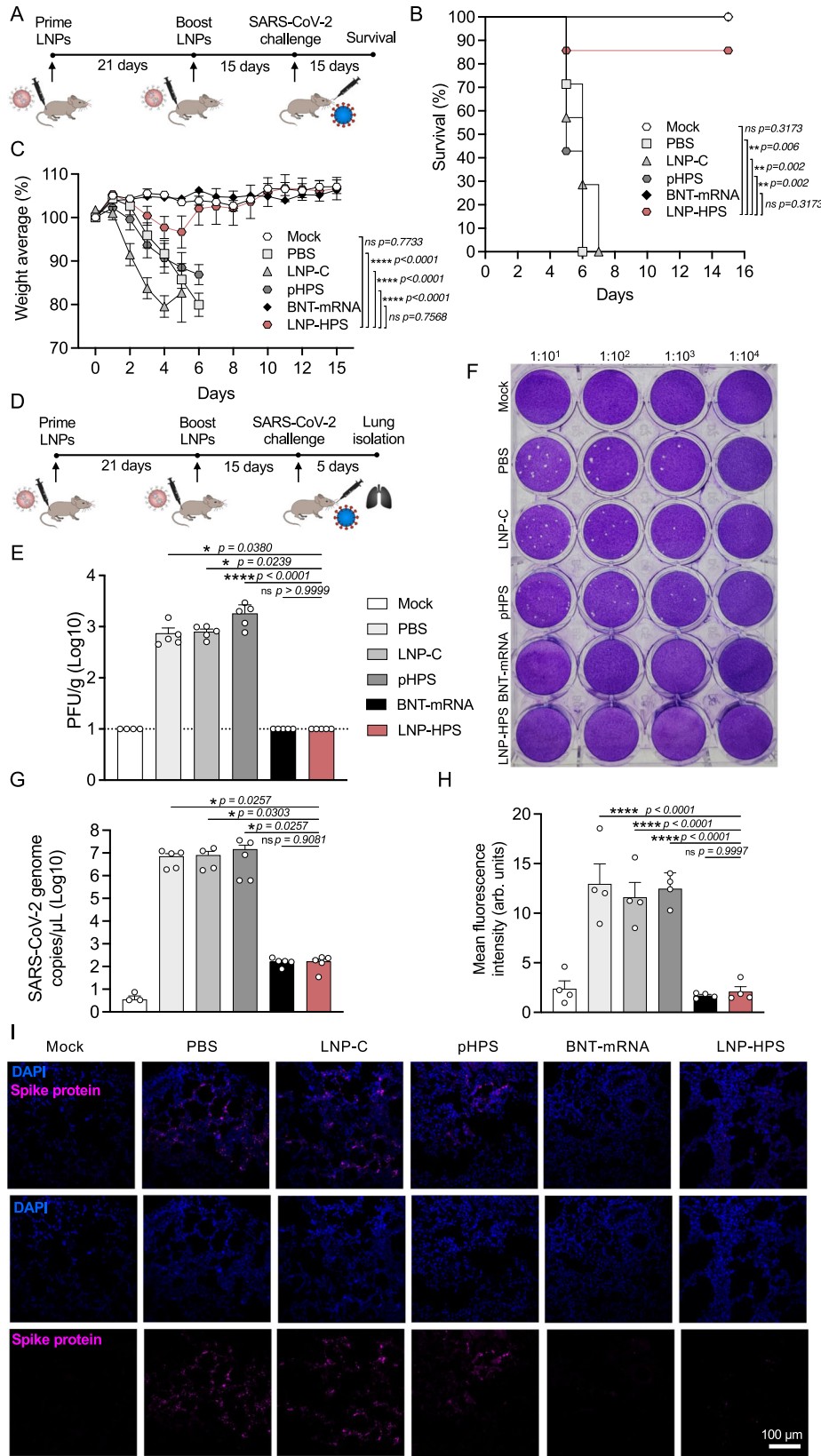

was evident edema, thickened alveolar walls, and perivascular infiltrates. In contrast, hamsters immunized with LNP-HPS or BNT-mRNA demonstrated an overall preservation of pulmonary architecture. Notably, there was a less diffuse interstitial pneumonia, characterized by a moderate inflammatory infiltrate around the bronchi and discreet infiltrates around the vessels and in the alveoli (Fig. 8G–J). Similar to the observations in K18-hACE2 mice, no histological changes or damage were detected in other tissues of hamsters, including the brain, kidney, heart, and liver, for all groups compared to the Mock group (Supplementary Fig. 8).

**Fig. 4 | LNP-based DNA vaccine protects K18-hACE2 mice against SARS-CoV-2 variants Gamma lineage (P.1). A** Scheme of immunization: K18-hACE2 mice were immunized with controls and LNP-HPS and boosted with equivalent doses 3 weeks later. Immunized mice were challenged with lethal $6 \times 10^4$ PFU of SARS-CoV-2 variants Gamma lineage (P.1) 15 days after boost. **B**, **C** The lethality and weight loss were monitored for 15 days after challenge ($n = 7$/group). **D** Scheme of immunization wherein the lungs were harvested 5 days post infection for all immunized groups. **E**, **F** The plaque-forming unit measures the viable Vero cells treated with serum of immunized mice 5 days post infection ($n = 5$/group). Dashed line denotes limit of detection. **G** SARS-CoV-2 genome copies detected in the lung of the immunized mice at 5 days post infection (Mock $n = 3$; PBS, LNP-C, pHPS, BNT-

mRNA, and LNP-HPS $n = 5$). **H** Analysis of infection via spike protein in lung sections of immunized mice at 5 days post infection via immunofluorescence from at least 20 fields of 3 different sections ($n = 5$/group). **I** Representative fluorescence images of lungs marked with anti-Spike antibody. Samples were stained with anti-SARS-CoV-2 Spike protein (RBD) antibody (magenta) and DAPI (blue) and acquired using a ×20 objective ($n = 5$/group). Data are presented as mean ± SEM; ns not significant, *$p < 0.05$, **$p < 0.01$, ****$p < 0.0001$. **B** Log-rank (Mantel–Cox) test. **C** Two-tailed, unpaired Spearman correlation to test. **E**, **H** One-way ANOVA with Dunnet's post hoc test. **G** Kruskal–Wallis followed by Dunn's multiple comparisons test. Source data are provided as a Source Data file.

## Humoral response in Syrian Hamster

To assess the humoral response elicited by immunization with LNP-HPS in larger animal model, we measured antigen-specific IgG titers in the sera of immunized hamsters before and after infection with SARS-CoV-2 variants P.1 and Omicron. All immunized groups were euthanized 5 days post infection (Fig. 9A). We found significantly enhanced anti-S IgG titers in serially diluted sera pre and post infection (1:200-1:6460) of hamster immunized with LNP-HPS or BNT-mRNA compared to LNP-C, pHPS and PBS control groups (Fig. 9B, C). Subsequently, we assessed the levels of neutralizing antibodies in the sera of immunized hamsters pre and post infection against SARS-CoV-2 variants P.1 and Omicron. The neutralization response induced by immunization with LNP-HPS or BNT-mRNA prior to infection against P.1 (PRNT-50% titer = 1:1240) (Fig. 9D, E) and Omicron (PRNT-50% titer = 1:1240) (Fig. 9H, I) was significantly higher compared to LNP-C, pHPS and PBS control groups. Notably, no significant difference was observed between the neutralization responses induced by LNP-HPS and BNT-mRNA groups. Similar results were observed in post infection sera neutralizing titers against P.1 (PRNT-50% titer = 1:1240) (Fig. 9F, G) and Omicron (PRNT-50% titer = 1:1240) (Fig. 9J, K) with both LNP-HPS and BNT-mRNA groups exhibiting higher P.1 and Omicron neutralization compared to LNP-C, pHPS, and PBS control groups. The results obtained in hamsters collectively support observed immunogenicity of LNP-HPS in K18-hACE2 mice and its comparable humoral response to that induced by BNT-mRNA vaccination.

## Discussion

Several technologies used to develop vaccines against SARS-CoV-2 worldwide have shown high protective efficacy[3,7]. Nevertheless, the protective efficacy of these vaccines seems to decrease against SARS-CoV-2 VOCs[26,27]. In addition, further research is still needed to assess the long-lasting protective immunity against SARS-CoV-2 VOCs. Here, we developed an LNP-based pDNA vaccine platform against SARS-CoV-2 VOCs. Compared to mRNA vaccines, DNA is cost-effective, easy to produce, and exhibits higher stability[12,14]. Plasmid encoding recombinant Hexapro spike protein was encapsulated in an optimized LNP formulation. Hexapro is a recombinant spike protein, derived from the SARS-CoV-2 wild-type, with additional six-proline substitutions (F817P, A892P, A899P, A942P), which exhibited higher expression than two-proline substitutions[28]. The two-proline substitution in the S2 subunit was used in the mRNA sequences of RNA vaccines approved[29].

To determine the lead LNP to be used in this study, we performed a high-throughput in vivo screening of a library of 15 LNPs encapsulating barcoded DNA (b-DNA). The b-DNA has been widely utilized to track the delivery of nucleic acids and enable hundreds of LNP formulations to be evaluated simultaneously in a single animal[19,30]. The in vivo delivery screen has an important role to identify potential LNP formulations for further assessment with therapeutic nucleic acid, such as pDNA[19,30]. This library of LNPs was composed of 15 formulations, containing each an ionizable lipid, which has been widely used for mRNA delivery[31], helper lipids, cholesterol, and a lipid-PEG, in varying combinations (Fig. 1A). Among these four components, we varied the cholesterol molar composition, and helper lipids, such as

DOPE, DOTAP, and DSPC as well as their molar composition based on previous studies that revealed changes in organ specificity of LNP formulations by varying helper lipid type and molar composition[32,33]. Beyond mRNA, previous studies demonstrated that ionizable lipids are also able to enhance DNA delivery in vitro and in vivo[19,22]. LNPs encapsulating distinct b-DNAs were administered intramuscularly as a single pool and the DNA from different tissues was isolated to obtain barcode counts via deep sequencing. As a result, we identified the top-performing LNPs with enhanced DNA delivery in spleen and lymph node, which were LNPs B2, B3, B4 and B10. Beyond specificity, safety concerns were reported recently, related to rare cases of myocarditis and pericarditis after immunization with LNP-based mRNA vaccines from Moderna and Pfizer-BioNTech[34,35]. Therefore, in our approach, we specifically selected top-performing LNPs B4 and B10 with enhanced b-DNA delivery to the spleen and lymph node, while demonstrating minimal delivery to the heart and liver (Fig. 1D and Supplementary Fig. 2).

To further validate our screening, we chose LNP B4 to assess the delivery of GFP pDNA in a dose response in antigen-presenting cells (macrophages, dendritic cells and B cells), fibroblasts, myoblasts and endothelial cells (Fig. 2A and Supplementary Fig. 3A), all relevant cell lines for intramuscular administration. Treatment with LNP B4 resulted in significantly higher GFP expression than control for all cell lines at all doses, indicating enhanced pDNA delivery. Our results revealed LNP B4 induced higher GFP expression in antigen-presenting cells, fibroblasts and endothelial cells than LNP B10 (Fig. 2E, I, M, Q, and Supplementary Fig. 3E).

Next, the lead LNP formulation was formulated with pDNA plasmid Hexapro via microfluidic mixing to form LNP-HPS (Fig. 3A). Characterization revealed that our LNPs were monodisperse with a dense core and spherical morphology with a hydrodynamic diameter of approximately 150 nm and similar pKa to our previous studies[19,22]. We then decided to investigate whether the in vitro treatment of HEK-293 cells with LNP-HPS induced spike protein expression. The results were confirmed through immunofluorescence (Fig. 3G–H), indicating the potential of our LNP-HPS. Next, to assess the protective efficacy of our LNP-based DNA vaccine, K18-hACE2 transgenic mice were intramuscularly immunized with two doses of LNP-HPS, BNT-mRNA, LNP-C, pHPS or PBS and then challenged with SARS-CoV-2 VOC Gamma lineage (P.1) 15 days after boost. P.1 emerged as a predominant VOC in Brazil in 2021 with higher transmissibility rate and immune evasion ability[36]. While Omicron currently holds significance as a VOC, this variant induces milder disease in mice and hamsters[37,38]. In contrast, the Gamma lineage (P.1) has exhibited heightened virulence, pathogenicity, and lethality in K18-hACE2 mice compared to other variants[39]. Consequently, we opted for the Gamma lineage (P.1) to evaluate the protective efficacy of LNP-HPS vaccination. Additionally, we examined the neutralizing titers against Omicron using sera from immunized mice both before and after infection. Importantly, immunization with LNP-HPS or BNT-mRNA promoted robust protection against SARS-CoV-2 VOC Gamma lineage (P.1), with improved preclinical outcome and significantly reduced lethality compared to control groups (Fig. 4B, C). Also, the remarkable resistance against variant P.1 was

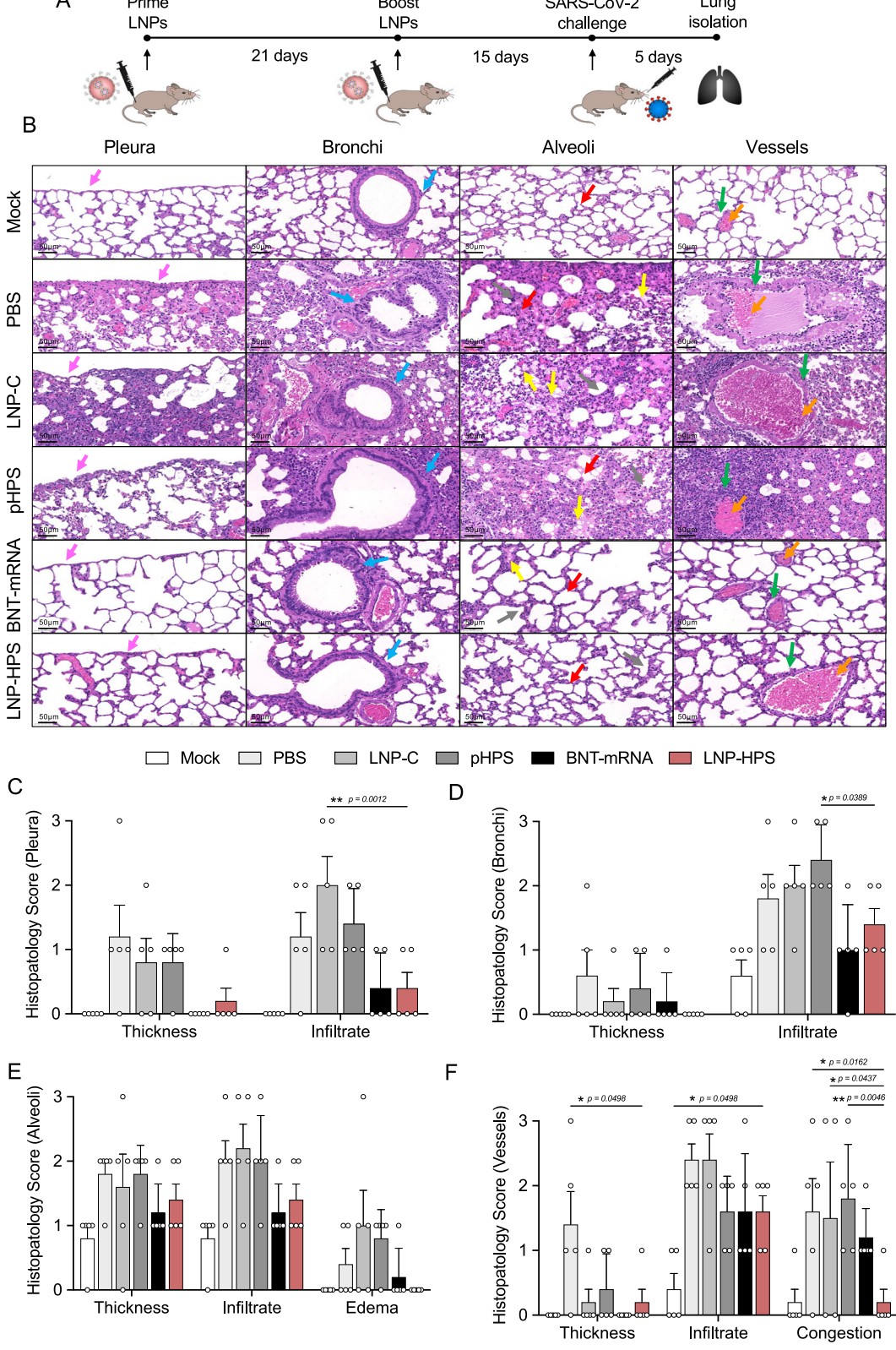

**Fig. 5 | LNP-HPS vaccine reduced lung damage. A** Scheme of immunization. K18-hACE2 mice were immunized with controls and LNP-HPS and boosted with equivalent doses 3 weeks later. Immunized mice were challenged with lethal $6 \times 10^4$ PFU of SARS-CoV-2 variants Gamma lineage (P.1). Lungs were harvested at 5 days post infection for all immunized groups for histopathological analysis. **B** Histopathological analysis at ×20 magnification of the lungs at 5 days post infection with the P.1 strain ($n = 5$/group). Histopathological score for **C** pleura, **D** bronchi, **E** alveoli, and **F** vessels ($n = 5$/group). The arrows indicate pleura (pink), bronchi (blue), alveoli (red), edema (yellow), vessels (green) and vessels congestion (orange). Data are presented as mean ± SEM; ns not significant, *$p < 0.05$, **$p < 0.01$, ***$p < 0.001$. Two-way ANOVA followed by Tukey's multiple comparison test. Source data are provided as a Source Data file.

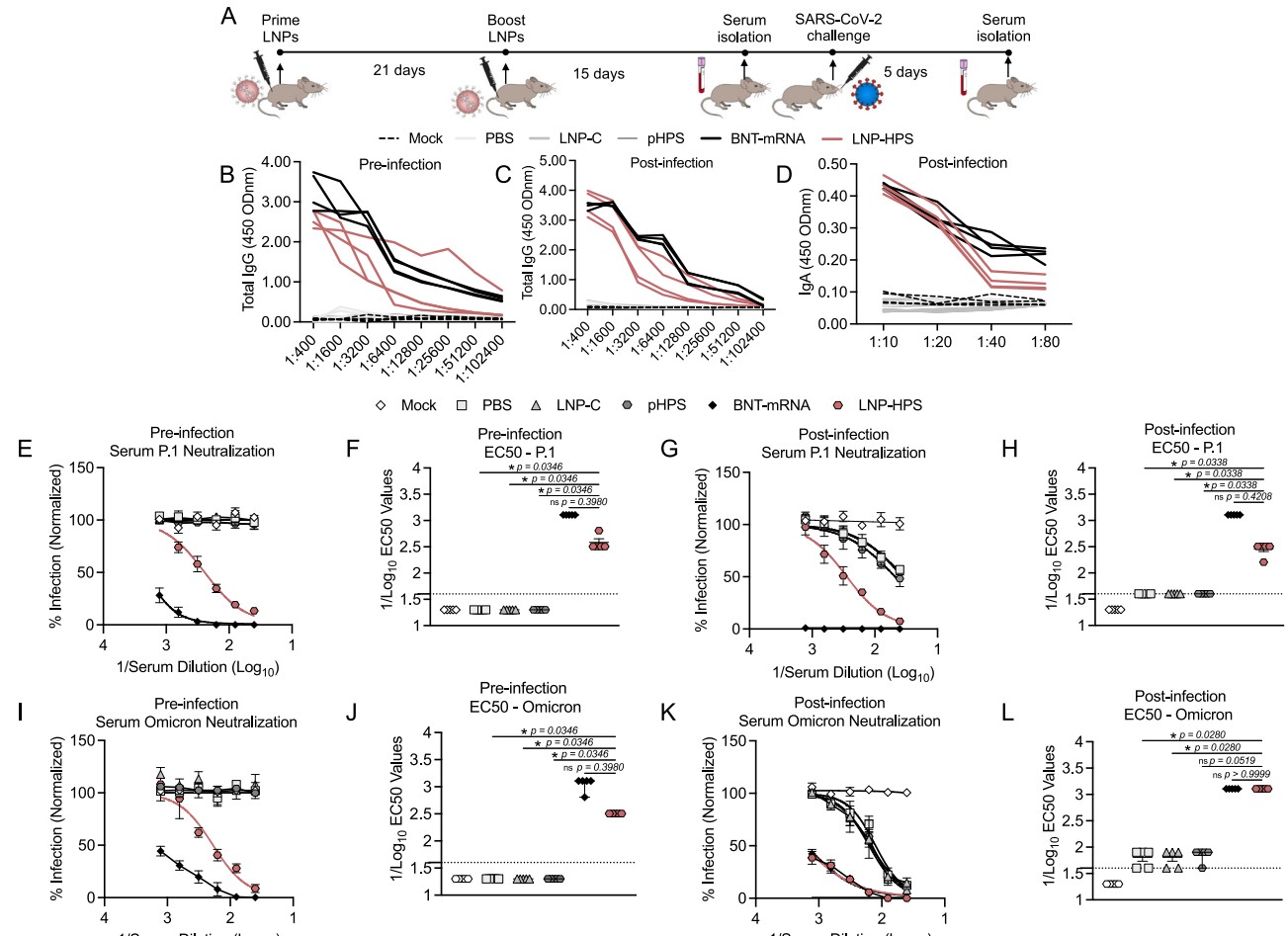

**Fig. 6 | LNP-based DNA vaccine-induced potent immune response and protection against SARS-CoV-2 infection and its variants. A** Scheme of immunization. K18-hACE2 mice were immunized with controls and LNP-HPS, and boosted with equivalent doses 3 weeks later. Immunized mice were challenged with lethal $6 \times 10^4$ PFU of SARS-CoV-2 variants Gamma lineage (P.1), sera were collected at 24 h before the infection (pre-infection), and 5 days post infection for all immunized groups and the lungs were collected 5 days post infection. Levels of total antigen-specific IgG anti-spike serum of mice at **B** pre-infection, and **C** 5 days post infection ($n = 4$/group). **D** Levels of total IgA anti-spike in lung homogenate at 5 days post infection ($n = 4$/group). Levels of neutralizing antibodies in sera of immunized mice at **E, F** pre-infection, and **G, H** 5 days post infection against P.1 ($n = 5$/group). Levels of neutralizing antibodies in sera of immunized mice at **I, J** pre-infection, and **K, L** 5 days post infection against Omicron ($n = 5$/group). Data are presented as mean ± SEM; ns not significant, *$p < 0.05$. **F, H, J, L** Kruskal–Wallis followed by Dunn's multiple comparisons test. Source data are provided as a Source Data file.

confirmed by the reduced viral load in the lung of mice immunized with LNP-HPS or BNT-mRNA, as determined via RT-qPCR, plaque assays, and immunofluorescence (Fig. 4E–I) in mice and hamsters. Additionally, mice and hamsters immunized with LNP-HPS or BNT-mRNA also showed significantly reduced lung pathology as well as lung inflammation and diffuse interstitial pneumonia (Fig. 5B–F). The ionizable lipid component is reported as crucial for adjuvant activity of LNPs, including for enhanced humoral and T cell response[40]. Intramuscular vaccination with LNP-HPS and BNT-mRNA induced very high titers of antigen-specific IgG in sera before and after infection with P.1 and Omicron. In addition, enhanced IgA titers in lung homogenate (Fig. 6B–D), suggests the role of IgA in the early stage of neutralizing response to SARS-CoV-2[41]. Omicron is considered the current dominant VOC due to its high transmissibility and immune evasion, which may affect the efficacy of current vaccines[42,43]. Interestingly, immunization with LNP-HPS or BNT-mRNA were able to induce neutralizing titers pre and post infection not only for P.1 but also for Omicron (Fig. 6I–L). In contrast, it has been reported that humoral responses of approved vaccines were less effective against some VOC, such as P.1 and Omicron[42,44,45]. Reduced neutralizing antibodies against P.1 and Omicron variants were reported in sera of individuals immunized with mRNA vaccines, which supported updated strategies and new vaccine

platforms[42,45]. It should be noted that a booster vaccination with BNT162b2 or mRNA-1273 mRNA vaccines were able to induce Omicron neutralizing antibodies[46]. Much of the discussion about the efficacy of nucleic acid vaccines revolves around cellular immune response[47]. The CD8+ and CD4+ T cell responses were effective for approved vaccines against VOCs[48]. In our study, vaccination with LNP-HPS or BNT-mRNA induced robust RBD-S1 Spike-specific CD8+ and CD4+ T cell responses, accompanied by upregulation of Gran-B and IFN-ɣ in memory CD8+ and CD4+ T subsets (Fig. 7). The induction of CD8+ and CD4+ T cell responses by the LNP-HPS or BNT-mRNA vaccine is consistent with reports of other nucleic acid vaccines[12,48,49]. Furthermore, vaccination with LNP-HPS increased central and effector/effector memory CD8+ and CD4+ T cell subpopulations, with upregulation of Gran-B and IFN-ɣ in memory CD8+ and CD4+ T cells (Fig. 7). Importantly, our findings reveal that protective efficacy and immunogenicity of LNP-HPS against P.1 is comparable to that elicited by BNT-mRNA vaccination.

In 2021, Zydus Cadila's DNA vaccine for SARS-Cov-2, known as Zydus-D[15,49], was approved for emergency use in India. Additionally, INO-4800 from Inovio Pharmaceuticals, is in phase III clinical trials[50]. However, both are intradermally administered using PharmaJet Tropis needle-free injection or electroporation. In contrast to these DNA vaccines, our approach utilizes nanoparticles to deliver DNA, which

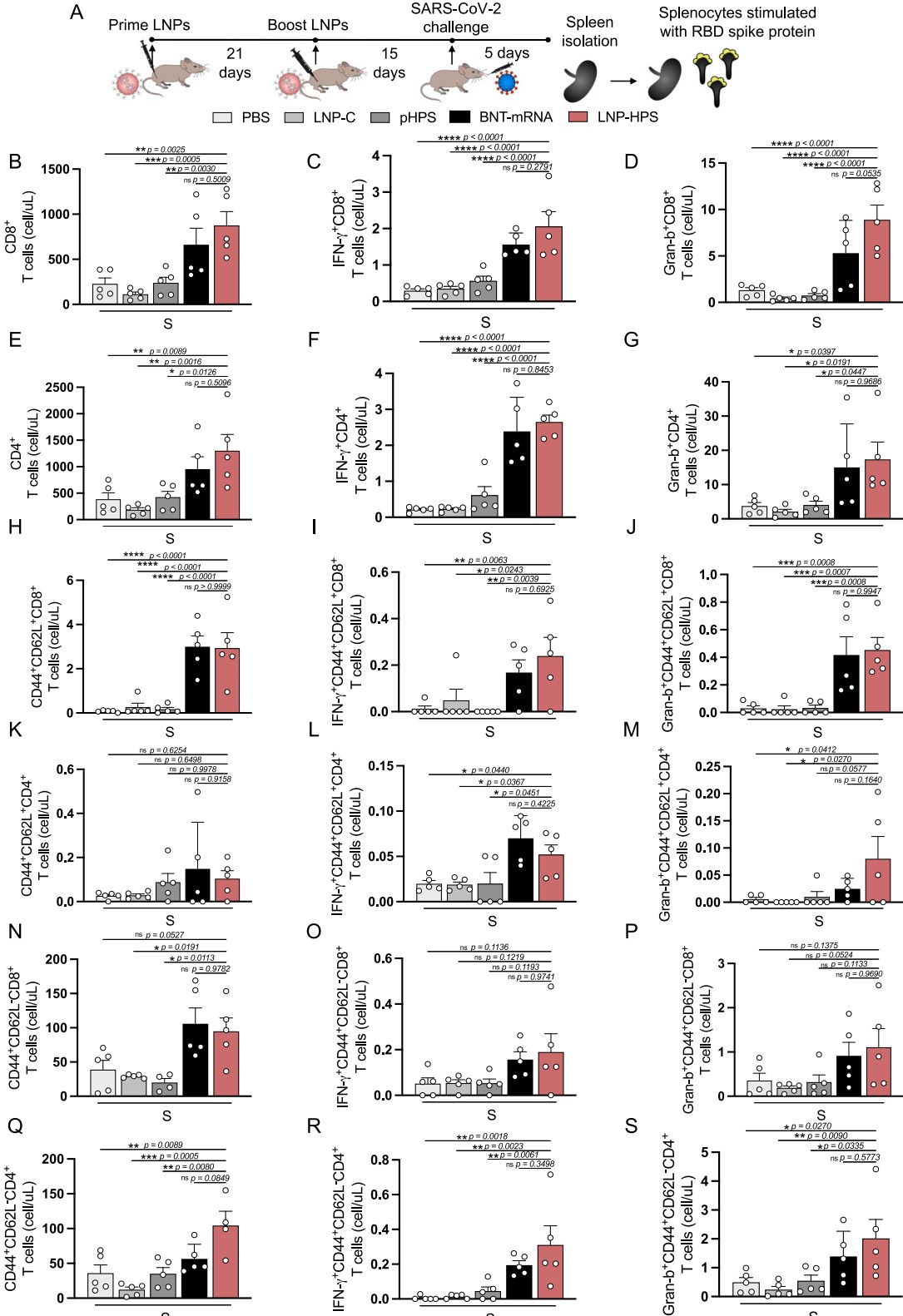

**Fig. 7 | LNP-based DNA vaccine-induced antigen-specific T cell response.**
**A** Scheme of immunization. K18-hACE2 mice were immunized with controls and
LNP-HPS and boosted with equivalent doses 3 weeks later. Immunized mice were
challenged with lethal $6 \times 10^4$ PFU of SARS-CoV-2 variants Gamma lineage (P.1).
Spleens were harvested 5 days post infection for all immunized groups and sple-
nocytes were stimulated with RBD-S1 protein. The graph shows cell number of
**B** CD8+ and **E** CD4+ T cells producers of **C**, **F** IFN-γ and **D**, **G** Gran-b ($n = 5$ samples/
group). Cell number of central memory and effector/effector memory from
**H**, **N** CD8+ and **K**, **Q** CD4+ T cells; and **I**, **L**, **O**, **R** IFN-γ and **J**, **M**, **P**, **S** Gran-b production
from memory subsets ($n = 5$ samples/group). Data are presented as mean ± SEM; ns
not significant, *$p < 0.05$, **$p < 0.01$, ***$p < 0.001$, ****$p < 0.0001$. One-way ANOVA
followed by Tukey's multiple comparison test. Source data are provided as a Source
Data file.

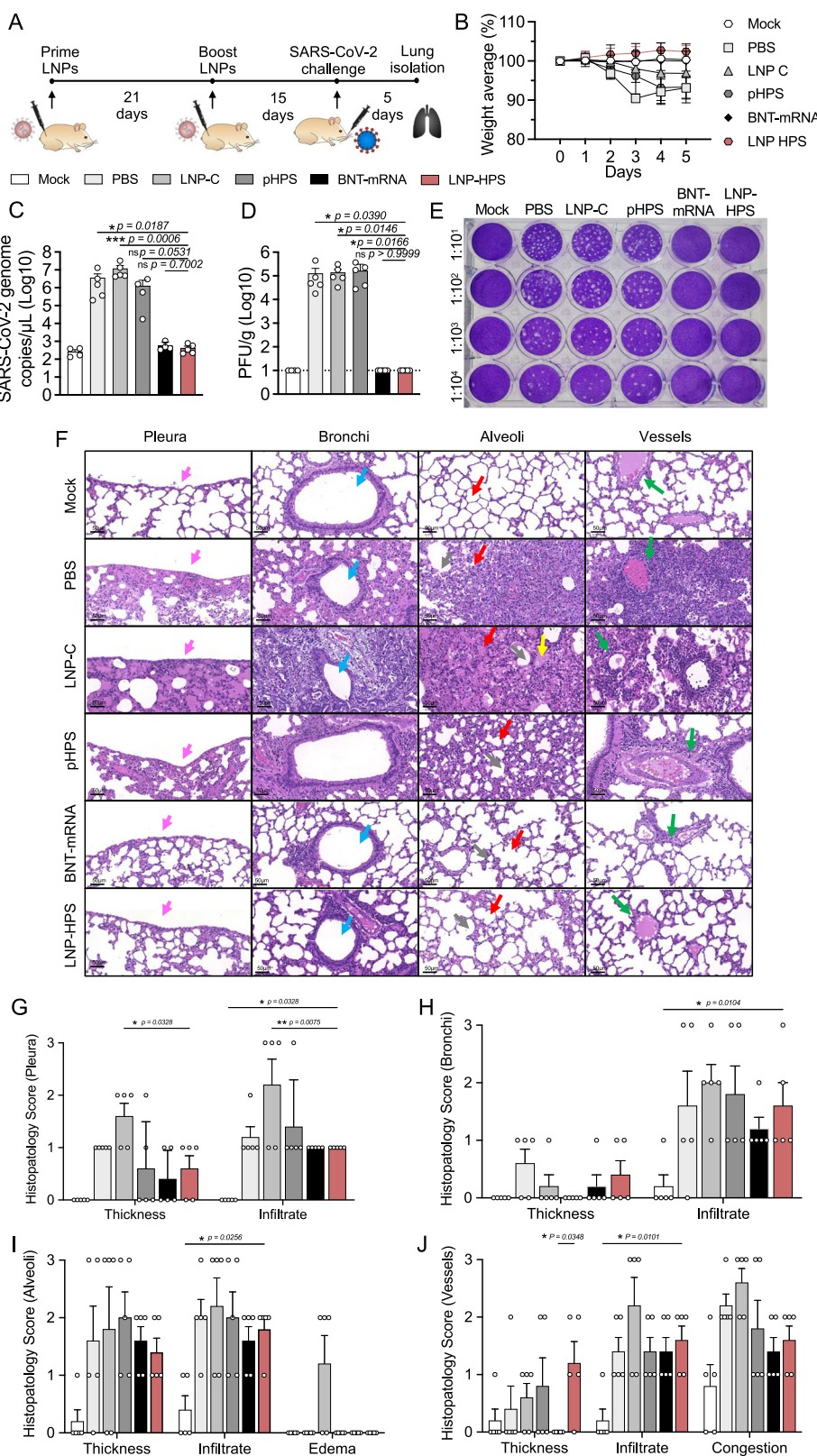

offers several advantages, including no need for sophisticated equipment for intramuscular injection and targeted delivery to specific cells, such as antigen-presenting cells. Both DNA vaccines have shown effective humoral and T cell response against SARS-CoV-2, with no safety concerns. In another preclinical study, 5 mg of DNA vaccine administered via intramuscular route in two doses without adjuvant induced protective humoral and cellular immune in rhesus macaques[51]. Similarly, naked pDNA intramuscularly administered without adjuvant induced antigen-specific IgG response and neutralizing antibodies against SARS-CoV-2 in mice, rabbits, and rhesus macaques[52]. In another study, it was demonstrated that a DNA vaccine injected intradermally using a pyro-drive jet injector was able to induce higher levels of neutralizing antibodies than intramuscular injection[53]. In addition, it was reported that an intramuscular injection of DNA

**Fig. 8 | Immunogenicity and efficacy of LNP-based DNA vaccine against SARS-CoV-2 variants Gamma lineage (P.1) in hamsters. A** Scheme of immunization: Syrian hamsters were immunized with controls and LNP-HPS and boosted with equivalent doses 3 weeks later. Immunized hamsters were challenged with $6 \times 10^5$ PFU of SARS-CoV-2 variants Gamma lineage (P.1) 15 days after boost. Lungs were harvested 5 days post infection for all immunized groups. **B** The weight loss was monitored for 5 days after challenge ($n = 5$/group). **C** SARS-CoV-2 genome copies detected in the lung of the immunized hamsters at 5 days post infection (Mock, pHPS, and BNT-mRNA $n = 4$; PBS, LNP-C, and LNP-HPS $n = 5$). **D**, **E** The plaque-forming unit measures the viable Vero cells treated with serum of immunized

hamsters 5 days post infection ($n = 5$/group). Dashed line indicates limit of detection. **F** Histopathological analysis at ×20 magnification of the lungs at 5 days post infection with the P.1 strain ($n = 5$/group). Histopathological score for **G** pleura, **H** bronchi, **I** alveoli, and **J** vessels ($n = 5$/group). The arrows indicate pleura (pink), bronchi (blue), alveoli (red), edema (yellow), vessels (green) and vessels congestion (orange). Data are presented as mean ± SEM; ns not significant, *$p < 0.05$, **$p < 0.01$, ***$p < 0.001$. **B** Two-tailed, unpaired Spearman correlation to test. **C** Kruskal–Wallis followed by Dunn's multiple comparisons test. **D** One-way ANOVA with Dunnet's post hoc test. **G**–**J** Two-way ANOVA followed by Tukey's multiple comparison test. Source data are provided as a Source Data file.

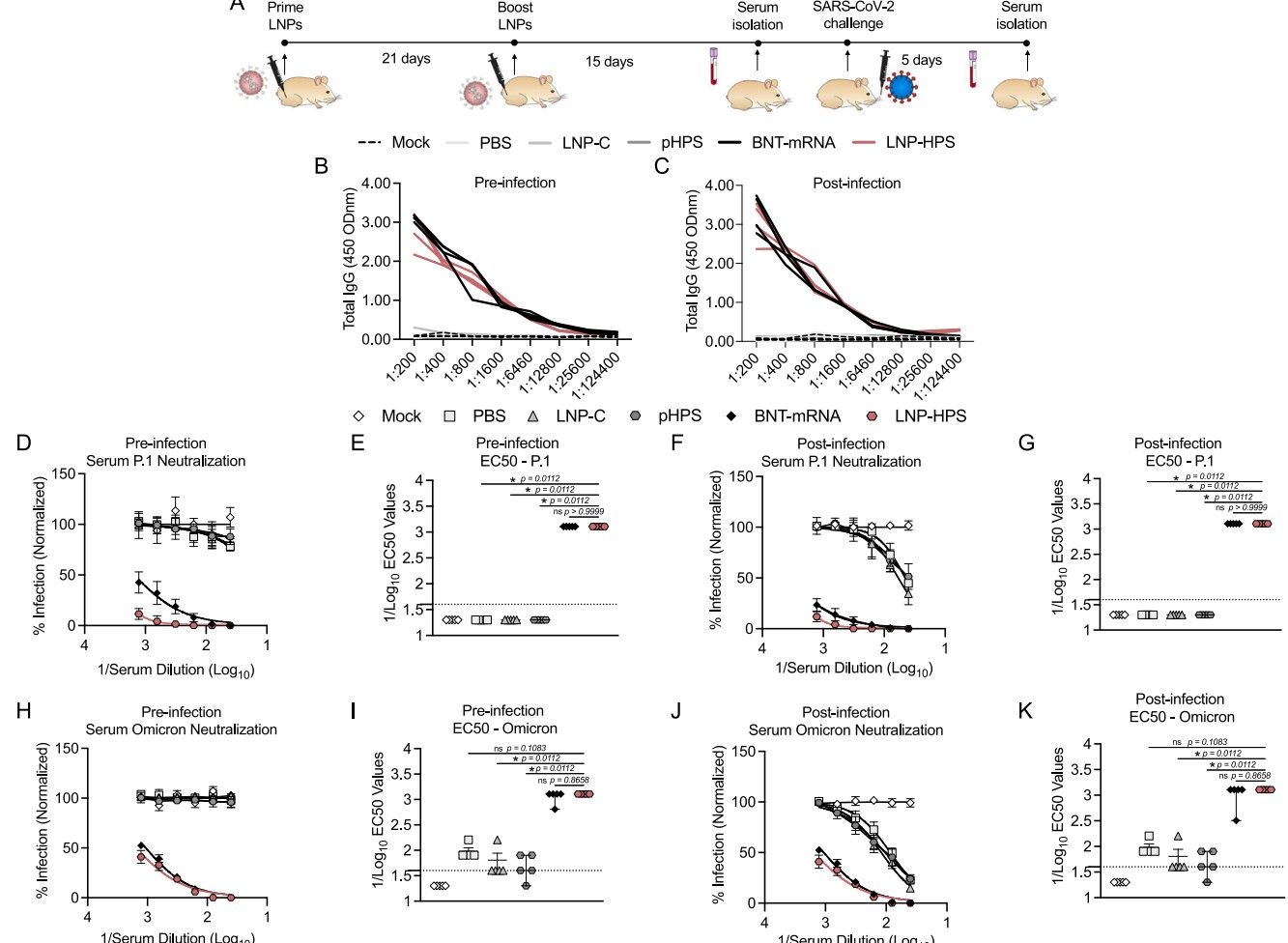

**Fig. 9 | LNP-based DNA vaccine-induced potent immune response against SARS-CoV-2 infection and its variants in hamsters. A** Scheme of immunization. Syrian hamsters were immunized with controls and LNP-HPS, and boosted with equivalent doses 3 weeks later. Immunized mice were challenged with $6 \times 10^5$ PFU of SARS-CoV-2 variants Gamma lineage (P.1), sera was collected at 24 h before the infection and 5 days post infection for all immunized groups. Levels of total antigen-specific IgG anti-spike serum of hamsters **B** at pre-infection, and **C** 5 days

post infection ($n = 4$/group). Levels of neutralizing antibodies in sera of immunized hamsters at **D**, **E** pre-infection, and **F**, **G** 5 days post infection against P.1 ($n = 5$/group). Levels of neutralizing antibodies in sera of immunized hamsters at **H**, **I** pre-infection, and **J**, **K** 5 days post infection against Omicron ($n = 5$/group). Data are presented as mean ± SEM; ns not significant, *$p < 0.05$. **E**, **I**, **G**, **K** Kruskal–Wallis followed by Dunn's multiple comparisons test. Source data are provided as a Source Data file.

vaccine with alum adjuvant induced T cell response and neutralizing antibodies in hamsters infected with SARS-CoV-2[54]. Taken together, these data promise effectiveness and safety of DNA vaccine platforms against SARS-CoV-2 and VOCs.

To our knowledge, no DNA vaccines against SARS-CoV-2 use our delivery approach, based on LNPs formed by ionizable lipids, which act as an adjuvant. Indeed, coordinated response among neutralizing antibodies, CD4+ T cells, and CD8+ T cells is crucial for a higher vaccine efficacy, which is induced by immunization with LNP-HPS. Beyond efficacy, LNP-based DNA vaccines have key advantages, including

lower costs, ease of production at large scales, thermostability, and rapid manufacturing.

In conclusion, we report that high-throughput in vivo screening using b-DNA can be used to identify lead LNP formulations as well as parameters to design optimized LNPs for pDNA delivery in spleen after intramuscular administration. In addition, we validated our developed LNP-based pDNA vaccine against SARS-CoV-2 VOCs P.1 and Omicron. We demonstrated that immunization with LNP-HPS promoted robust protection against P.1, with improved preclinical outcome as well as significantly decreased lethality, viral load and consequently reduced lung

damage. Importantly, LNP-HPS vaccine-induced robust CD4+ and CD8+ T cell response, including memory T cells, and was effective to induce neutralizing antibodies titers for both P.1 and Omicron VOCs. Finally, the protective efficacy and immunogenicity elicited by vaccination with LNP-HPS is comparable to BNT-mRNA. Collectively, our results demonstrated the potential of our LNP-HPS against SARS-CoV-2 VOCs

## Methods

### Plasmid DNA (pDNA)

*E. coli* Stbl3 transformed with pZsGreen-N1 (Clontech laboratories, no. 632448) encoding ZsGreen, a green fluorescent protein (GFP), or recombinant HexaPro Spike plasmid (Addgene, no. 154754) (Supplementary Fig. 2) were grown in LB medium (Sigma-Aldrich, no. L3022) containing kanamycin (50 μg/ml) or ampicillin (100 μg/mL), respectively, at 37 °C for 16 h. The plasmids were purified by PureLink™ HiPure Plasmid Maxiprep Kit (Invitrogen, no. K210007), following the manufacturer's instructions, and quantified using a NanoDrop-OneC-UV-Vis (ThermoFisher, no. ND-ONE-W).

### Preparation of LNPs

Nanoparticles were prepared using a 10:1 weight ratio of ionizable lipid to nucleic acid. An ethanol phase consisting of ionizable lipid, cholesterol (Avanti Polar Lipids, no. 700100 P), 1,2-dimyristoyl-sn-glycero-3-phosphoethanolamine-N-[methoxy(polyethyleneglycol)-2000] (ammonium salt) (C14-PEG 2000, Avanti Polar Lipids, no. 880150 P), and optionally 1,2-dioleoyl-sn-glycero-3-phosphoethanolamine (DOPE, Lipoid), 1,2-distearoyl-sn-glycero-3-phosphocholine (DSPC, Lipoid), or 1,2-dioleoyl-3-trimethylammonium-propane (DOTAP, Lipoid) at varying molar ratios (Supplementary Table 1) was combined with an aqueous phase containing pDNA in 10 mM citrate buffer, pH 5, to form LNPs via microfluidic mixing at a volume ratio of 3:1. LNPs were dialyzed against 1× PBS for 2 hours using dialysis cassettes with a molecular weight cutoff of 20 kDa (Thermo Scientific, no. 66003), sterile filtered using a 0.22 μm filter, and stored at 4 °C for later use.

### Nanoparticle characterization

pDNA nanoparticles were characterized by hydrodynamic diameter (HD), polydispersity index (PDI), and zeta potential using a Zetasizer Nano ZS machine (DLS, Malvern Panalytical). For analysis of LNP structure using cryogenic transmission electron microscopy (cryo-TEM), LNP samples were prepared in a vitrification system (25 °C, -100% humidity). Vitrified samples were examined using Tecnai G2-20 - FEI SuperTwin 200 kV at the Microscopy Center of UFMG. pDNA concentration in LNPs for in vitro and in vivo use was quantified using a NanoDrop Spectrophotometer (ThermoFisher), and Qubit™ 1× dsDNA HS Assay Kit (ThermoFisher, no. Q33230) in Qubit equipment (ThermoFisher) according to manufacturer's instructions.

### *LNP* pKa measurements

To calculate the pKa of each LNP, the 2-(p-toluidinyl)naphthalene-6-sulfonic acid (TNS) (Sigma-Aldrich, no. T9792) assay was performed. Briefly, a buffer solution was prepared after combining 150 mM sodium chloride, 20 mM sodium phosphate, 25 mM sodium citrate, and 20 mM sodium acetate. The pH was adjusted to 2 to 12 in increments of 0.5. Next, 140 μL of each pH-adjusted solution and 5 μL of each LNP were added in triplicate in a 96-well plate. TNS was then added to each well for a final concentration of 6 μM. The fluorescence (ex/em 325/435 nm) was measured on Cytation 5 Cell Imaging (Biotek). From the emitted fluorescence we built a sigmoidal curve-fit to analyze and determine the 50% of protonation, at which point the pH value corresponds to the pKa.

### DNase I protection assay

To assess the protective properties of LNPs against enzymatic degradation of pDNA we performed the DNase I Protection Assay using a DNase I, RNase-free enzyme (Thermo Scientific, EN0521). LNP-encapsulated pDNA or naked pDNA at an amount of 1 μg were incubated with DNase I for 60 minutes at 37 °C. Naked pDNAs were used as positive control. The reaction volume was made up to 10 μL using ultrapure sterile water. Then, 5 μL EDTA was added and incubated for 60 minutes at 65 °C to stop the reaction. Following, isopropyl alcohol was added at 5:1 proportion (v/v) alcohol:LNP, when applied. Next, 2.5 μL (50 UI) of heparin was added in each sample and incubated for 60 minutes at room temperature. All samples were run by gel electrophoresis on 0.8% agarose stained with SYBR Safe in ×0.5 Tris-acetate-EDTA (TAE) buffer for 140 minutes. The gel was visualized and documented using Kodak Gel Logic 1500 Imaging System.

### Preparation of the b-DNA library

The b-DNA library was designed using a previously described protocol[30]. Briefly, each b-DNA consists of single-stranded DNA containing 61 nucleotides with 5 consecutive phosphorothioate bonds at each end. The barcode region (reading region) was composed of 10 nucleotides. An additional 10 random nucleotides were added at the 3' end of the barcode region to monitor PCR over-amplification. The 5' and 3' ends of each b-DNA were conserved and contained priming sites for Illumina adapters. A full list of b-DNA sequences can be found in Supplementary Table 2. All oligonucleotides in this study were synthesized and purified (standard desalting procedure) by Integrated DNA Technologies.

### Ethics statement

This study received approval from the Ethical Committees on the use of animals in research at the Federal University of Minas Gerais (UFMG). The experiments involving mice and hamsters adhered to institutional guidelines for animal ethics and were approved by the Institutional Ethics Committees at UFMG, specifically Commission on Animal Use (CEUA) 177/2020, 245/2021, and 165/2021, for C57BL/6 mice, K18-hACE2 transgenic mice, and Syrian hamsters, respectively.

### Mice, hamsters and viruses

Female C57BL/6 mice and female Syrian hamsters were purchased from Biotério Central at UFMG. Human Angiotensin Converting Enzyme transgenic mice (K18-hACE2) in the C57BL/6 background, originally from Jackson Laboratories, 8–10 weeks old, were bred at UFMG animal facilities and only female K18-hACE2 mice were utilized as a model of severe COVID-19. Female Syrian Hamsters, 8–10 weeks old, served as a model of mild COVID-19. The experiments were conducted in accordance with the recommendations in the Guide for the Care and Use of Laboratory Animals of the Brazilian National Council of Animal Experimentation (CONCEA). Infections of K18-hACE2 and Syrian hamsters were performed in the Animal Biosafety Level 3 (ABSL-3) facility at the Institute of Biological Sciences from UFMG. All animals were maintained with 12 h light/dark cycle with humidity of 50-58% and temperature of 25 °C. The SARS-CoV-2 viral strain used in this study belonged to the lineage P.1 (EPI_ISL_13017802) and Omicron (EPI_ISL_7699344) variants. Viral stocks were propagated in Vero CCL81 in a humidified incubator at 37 °C with 5% $CO_2$ and monitored for cytopathic effects (CPE) daily up to 72 h. Viruses were titrated in Vero CCL81 cells by plaque-forming units (PFU) assay, and viral aliquots were kept at −80 °C until further use.

### In vivo b-DNA delivery

To assess the LNP biodistribution in several tissues, we used different b-DNAs encapsulated in 15 LNPs with distinct components or their molar ratios (Supplementary Table 1). Briefly, the 15 LNPs were pooled and injected in 8-week-old female C57BL/6 mice (20−25 g) via intramuscular route (IM) through the quadriceps femoris muscle at the amount of 0.5 μg. Tissue samples (heart, liver, lung, spleen, draining

lymph nodes and muscle) were harvested 4 hours after administration, immediately frozen in liquid nitrogen, and stored in a −80 °C freezer for further analysis. The organs were then macerated using autoclaved surgical scissors and pistils in DNAse and RNAse-free microtubes. The DNA was extracted using the Purelink Genomic DNA mini-Kit (ThermoFisher, Cat. no. K1820-02). The purified DNA was quantified via spectrophotometry using Nanodrop Lite (ThermoFisher Scientific), and the purity was measured by A260/A280 ratio and gel electrophoresis.

### b-DNA amplification

After extraction of DNA from the tissues, the samples were then submitted to polymerase chain reaction (PCR). Initially, a stock solution of a primer mix that would be used was prepared, containing 5 μL of miseq universal at 5 μM, 0.5 μL of barcode base at 0.5 μM and 94.5 μL of Tris-EDTA (TE) pH 8.0. PCR was performed as follows: 10.3 μL of nuclease-free water, 0.75 μL of 100% DMSO, 0.5 μL of 10 mM DNTPs, 1.0 μL of MgCl$_2$, 1.0 μL of primer mix, 5 μL of DNA (10 ng/μL) and 1.2 μL of sequence primer at 5 μM (a unique primer sequence for each organ as shown in Supplementary Table 2). Cycling conditions were 98 °C for 48 seconds, 98 °C for 12 seconds, 67 °C for 22 seconds, 72 °C for 28 seconds and 72 °C for 5 minutes, repeated for 39 cycles. The PCR products were visualized by electrophoresis gel in 1.4% agarose, and the bands containing the DNA fragments of interest were excised and purified using the Purelink Quick Gel extraction Kit (ThermoFisher - Ref: K210012). The purified products were stored under refrigeration at −20 °C until sequencing.

### Deep sequencing and delivery quantification

The deep sequencing was performed using Illumina next-generation sequencing. Initially, a library was prepared using 5 μL of each sample of the DNA fragment purified from the gel, previously diluted to a final concentration of 4 nM. Afterwards, the libraries were sequenced by the Illumina Miseq platform on the Illumina sequencer at UFMG. The sequencing data was analyzed using an algorithm in Python, developed to quantify the number of reads of the data generated from each b-DNA in the analyzed tissues. The delivery of specific b-DNA to a particular organ of interest was calculated according to the following: First, the number of reads of a specific b-DNA in an organ was divided by the total number of b-DNA reads from the same organ for an assessment of the biodistribution of b-DNA in the same organ. This analysis allowed the identification of the lead LNP formulations for that specific organ.

### Cell culture

The macrophage cell line (RAW 264.7), fibroblast cell line (L929), endothelial cell line (bEnd.3), myoblasts cells (C2C12), and human embryonic kidney 293 (HEK-293) cells were cultured in Dulbecco's Modified Eagle Medium (DMEM) F12 medium supplemented with 10% v/v fetal bovine serum (FBS) and penicillin/Streptomycin (1% v/v). Dendritic cells (JAWSII) were cultured in humidified atmosphere at 37 °C and 5% CO$_2$ in Alpha minimum essential medium containing ribonucleosides, deoxyribonucleosides, 2 mM L-glutamine, 5 ng/mL murine GM-CSF, and 20% of fetal bovine serum. For experiments, cells were activated with LPS (2 μg/mL) from Escherichia coli for 24 h before transfection. The African green monkey kidney (Vero, subtype CCL81) was cultured in high-glucose DMEM supplemented with 10% v/v fetal bovine serum (FBS) and penicillin/Streptomycin (1% v/v). Cells were grown at 37° under 5% CO$_2$ in a humidified incubator.

### In vitro LNP transfection

To validate the b-DNA-LNP screening platform, we investigated the transfection efficiency and optimal dose for cell transfection of lead LNPs in different cell lines—a macrophage cell line (RAW 264.7), dendritic cell line (JAWSII), a fibroblast cell line (L929), myoblasts cell line

(C2C12) and endothelial cell line (bEnd.3). L929, C2C12 and bEnd.3 cells were plated with 5000 cells per well, whereas RAW 264.7 and JAWSII cells were plated with 10,000 cells per well in 96-well flat-bottom plates and incubated for 24 hours prior to treatment with LNPs. A serial dilution of each DNA-LNP formulation in PBS was prepared at doses varying from 0.00625 to 0.8 μg of DNA. The transfection efficiency was measured at 24, 48 and 72 h using Cytation 5 Cell Imaging (Biotek) with ×4 objective, and data was analyzed using a Python script, which allowed quantifying the fluorescence intensity in each sample. Cell viability testing was performed after 72 hours LNPs transfection with alamarBlue™ (ThermoFisher Scientific: no. DAL1025) assay, and the fluorescence was measured using ex/em 530/590 nm in Cytation 5 Cell Imaging (BioTek). To assess the transfection in B cells, spleens from C57BL/6 mice were harvested and macerated using a 70 μm pore cell strainer (Cell Strainer, BD Falcon) followed by erythrocyte lysis. Splenocytes were plated overnight using RPMI medium supplemented with 10% FBS, 2 mM L-glutamine, 50 μg/mL streptomycin, and 50 units/mL penicillin and then treated with B4 or B10 LNPs for 24 h. The gating strategy CD45 + CD3-CD19 + GFP+ was used to assess transfection in B cells treated with B4 or B10 via flow cytometry.

### In vitro pDNA expression

To investigate the transfection efficiency of lead LNP-encapsulated recombinant HexaPro spike plasmid in vitro, HEK-293 cells were treated with control LNP or LNP-HPS (0.1 μg of DNA per well) for 48 h. For the immunofluorescence assay, cells were fixed with 4% paraformaldehyde (PFA). Cells were blocked to prevent nonspecific binding (PBS, 3% BSA, 0.5% Triton) and then immunolabeled with SARS-CoV-2 antibody (no:40591-t62, 2019-nCoV Spike RBD Antibody, Sino Biological) at a 1:500 dilution overnight. Cells were then incubated with specific secondary antibodies conjugated with goat anti-mouse Alexa Fluor 633 (1:1000, ThermoFisher, catalog no. A-21052) and 488-phalloidin (1:150, Molecular Probes, Thermo cat no. A-12379; Fisher Scientific; RRID:AB2315147) for 1 h at room temperature. The nuclei were stained with DAPI and images were acquired at the CAPI-UFMG using Cytation 5 Cell Imaging (Biotek) with a ×10 objective. The images were converted to 8-bit grayscale for fluorescence quantification and normalized by the non-transfected cells images. The mean pixel intensity was measured and plotted.

### Immunization in K18-hACE2 mice and Syrian hamsters

K18-hACE2 mice in the C57BL/6 background were used as a model to assess the immunogenicity and protective efficacy, while hamsters were used to assess the immunogenicity of LNP-HPS against severe COVID-19. Mice or hamsters were randomly divided into 6 groups ($n = 5$ animals/group): Mock (not-infected), PBS (as control), LNP-C (using control pDNA), pHPS, BNT-mRNA (Comirnaty® Original/Omicron BA.4-5−Biontech/Pfizer) and LNP-HPS (LNP encapsulating recombinant HexaPro Spike plasmid DNA). Mice were vaccinated via an intramuscular route with PBS or 22.5 μg of naked pHPS or encapsulated in LNP-HPS, 22.5 μg of control plasmid encapsulated in LNP-C or 1 μg of BNT-mRNA. Hamsters were vaccinated via an intramuscular route with PBS or 67.5 μg of naked pHPS or encapsulated in LNP-HPS, 67.5 of control plasmid encapsulated in LNP-C or 5 μg of BNT-mRNA. Three weeks after vaccination, all groups were boosted with the same dose. Blood samples were collected from all animals 36 days after prime. All experiments were independently performed twice. At week 5 after immunization, mice were challenged with 20 μL of saline or SARS-CoV-2 ($6 \times 10^4$ PFU, P.1) to create a lethal challenge model. Hamsters were challenged with 100 μL of saline or SARS-CoV-2 ($6 \times 10^5$ PFU, P.1). The animals were euthanized 5 days post-challenge. In mice, tissues were harvested to assess viral load, T cell response, disease parameters, production of inflammatory mediators (cytokines and chemokines) and histopathological or immunofluorescence analysis. In hamsters tissues were harvested to assess viral load, disease

parameters, and histopathological analysis. In both species blood was collected to assess IgG and serum-neutralizing antibodies. In a parallel experiment using the same groups ($n = 7$ mice/group) and vaccination doses, mice were challenged with 20 μL of saline or SARS-CoV-2 ($6 \times 10^4$ PFU, P.1) and monitored daily for 15 days by measuring body weight loss, clinical signs, and lethality rates.

## Viral load

To assess viral load, a serial dilution of lung homogenate of immunized mice or hamsters was incubated in monolayers of Vero CCL81 cells ($10^5$ cells/well) plated in 24-well plates for 1 h at 37 °C. Fresh semisolid medium containing 1.5% carboxymethylcellulose (CMC) was added, and the culture was maintained for 72 h at 37 °C. Cells were fixed with 10% formaldehyde for 2 h at room temperature and then stained with crystal violet (0.4%). The virus titers were determined by plaque-forming units (PFU) per milliliter. The viral load was also assessed via RT-qPCR as follows: molecular diagnosis for detection of SARS-CoV-2 (primer N1) was performed in accordance with the CDC 2019-Novel Coronavirus (2019-nCoV) Real-Time RT-PCR Diagnostic Panel using 2019-nCoV RUO kit (IDT) for N1 and N2 gene regions. Real-time PCR was performed using CFX Opus Real-Time PCR System (Biorad). The standard curve and negative controls were used to validate the method.

## Immunofluorescence assay

To investigate the viral load in lung tissue via immunofluorescence, the lungs of immunized and control groups were harvested to prepare cryosections of 5 μm thickness. Samples were stained with SARS-CoV-2 (1:500, no:40591-t62, 2019-nCoV Spike RBD Antibody, Sino Biological), and goat anti-rabbit Alexa Fluor 633 (1:100, ThermoFisher Scientific, no. A-21052) as the secondary antibody overnight at 4 °C. Nuclei were stained with DAPI (ThermoFisher) and mounted in Dako Fluorescence Mounting Medium (Dako, Santa Clara, CA) before images were taken using Zeiss LSM 880 Meta inverted confocal microscope (Oberkochen, Germany), with ×20 objective and a filter-based 633 nm channel at the Center for Acquisition and Processing of Images (CAPI-UFMG). The area marked with anti-Spike was quantified through a Python algorithm that converts the fluorescence image to an 8-bit grayscale image and obtains the mean pixel intensity.

## Histology

Histological features related to the injury caused by SARS-CoV-2 Gamma lineage (P.1) were analyzed in the lungs, brain, heart, kidney and spleen of immunized and control K18-hACE2 mice and hamsters. Harvested lungs were fixed with formaldehyde (4%), dehydrated, and embedded in paraffin to prepare sections of 5 μm thickness. The sections were stained with hematoxylin and eosin (H&E) for microphotograph analysis. The tissue morphological alterations observed in the lungs were determined using an inflammatory score system: (i) airway inflammation (up to 4 points), (ii) vascular inflammation (up to 4 points), (iii) parenchyma inflammation (up to 5 points), and general neutrophil infiltration (up to 5 points).

## ELISAs

To assess total IgG in the sera and IgA in lung homogenates, Nunc Maxisorp ELISA plates (ThermoFisher) were coated with 0.2 μg/well of SARS-CoV-2 S protein overnight. After being blocked with PBS-3% BSA for 2 h at 25 °C, plates were incubated with serial dilutions at 1:1 of heat-inactivated sera from immunized and control mice or hamsters in PBS supplemented with 1.5% BSA for 60 min at 37 °C. After 4 washes, plates were then incubated with anti-mouse IgG antibody at 1:2000 dilution or anti-hamster IgG antibody at 1:1000, for 60 min at 37 °C. To assess IgA content, lungs of mice were homogenized in PBS at a 1:2 ratio (w/v). Lung homogenates from immunized and control groups were serially diluted in PBS with 1.5% BSA. The samples were incubated for 60 min at

37 °C, washed 4 times and then incubated with anti-mouse IgA antibody at a 1:8000 dilution for 60 min at 37 °C. After 4 more washes, all plates were revealed with TMB substrate solution (ThermoFisher) for 15 min in the dark. Next, the reaction was stopped by adding 2 M $H_2SO_4$ and absorbance was immediately read at 450 nm. The results were expressed as raw optical density (OD).

## Neutralization test−plaque reduction neutralization test (PRNT)

Sera samples from mice and hamsters were heat-inactivated by incubation at 56 °C for 20 min in a dry bath. Serial dilutions of sera samples were mixed with an equal amount of virus suspension containing 100 plaque-forming units (PFU) in 0.1 mL. After incubation at 37 °C for 1 h, each virus-diluted serum sample (0.1 mL) was inoculated into one well of a 24-well plate containing a confluent monolayer of $10^5$ Vero cells. After incubation at 37 °C for 1 h, fresh semisolid medium containing 1.5% carboxymethylcellulose (CMC) was added, and the culture was maintained for 72 h at 37 °C. Cells were fixed with 10% formaldehyde for 2 h at room temperature and then stained with crystal violet (0.4%), and plaques were counted. The antibody titer was determined as the serum dilution that inhibited 50% of the tested virus inoculum (PRNT-50).

## Flow cytometry

To assess the T cell response, spleens from immunized mice were harvested 5 days post infection and macerated using a 70 μm pore cell strainer (Cell Strainer, BD Falcon) followed by erythrocyte lysis. Splenocytes were plated and stimulated overnight with 10 μg/mL of RBD S1 Spike protein. Phorbol 12-myristate 13-acetate (PMA) plus ionomycin (1 μg/mL) (Sigma, 25 ng/mL) was used as positive control. Next, stimulated splenocytes were cultivated for 4 h at 37 °C in RPMI supplemented with 10% FBS, 2 mM L-glutamine, 50 μg/mL streptomycin, and 50 units/mL penicillin, in the presence of Brefeldin A (ThermoFisher). Live/Dead (Invitrogen) marker was used to exclude dead cells. For extracellular staining, the following mAbs were used: anti-CD45 (Pacific Orange, 30-F11, ThermoFisher), anti-CD3 (Pacific Orange, 30-F11, ThermoFisher), anti-CD4 (PE-Cy7, GK1.5, ThermoFisher), anti-CD8 (eFluor 450, 53-6.7, ThermoFisher), anti-CD44 (SB600, IM7, ThermoFisher), anti-CD62L (PE-eFluor610, MEL-14, ThermoFisher). For intracellular antigens, cells were washed, fixed and permeabilized according to manufacturer's instructions (FoxP3 staining buffer set, eBioscience): anti-IFN-γ (APC-eFluor 780, XMG1.2, ThermoFisher) and anti-Gran-b (PE, NGZB, ThermoFisher). A BD LSR Fortessa was used for acquisition. CountBright™ Absolute Counting Beads were used for cell counting following the manufacturer's instructions (ThermoFisher). T cell subsets were gated on CD45+ live cells. In addition, serum of mice were harvested at 5 days post infection and kept at −80 °C until cytokine measurement by BD Cytometric Bead Array (CBA) Mouse Inflammation Kit (BD Ref. 552364). GraphPad Prism V10.0 (GraphPad software) and FlowJo V10.8.1 (BD) were used for data analysis and graphic presentation.

## Analysis of cytokines and chemokines

Total RNA extraction from lung samples (30 mg of tissue) was performed using PureLink RNA Mini Kit (Invitrogen), and RNA was quantified using Qubit (Qubit RNA HS Assay kit). The amount of 500 ng of total RNA/sample was submitted to cDNA synthesis with High-Capacity cDNA Reverse Transcription Kits (ThermoFisher Scientific) and stored at −20 °C. Expression levels of IL-1β, IL-6, IL-10, IL-17, TNF, CXCL14, CCL2, IFNβ, IFNγ and HPRT (as a reference gene) were evaluated by quantitative PCR using GoTaq Probe qPCR System (Promega). The specific set of primers and probes (Integrated DNA Technologies) predesigned for exon-exon junctions were used: Mm.PT.58.41616450, Mm.PT.58.10005566, Mm.PT.58.13531087, Mm.PT.58.6531092, Mm.PT.58.12575861, Mm.PT.58.21980826, Mm.PT.58.42151692, Mm.PT.58.30132453.g, Mm.PT.58.41769240,

and Mm.PT.39a.22214828. Real-time PCR was performed in a final volume of 10 μL on a CFX Opus Real-Time PCR System (Biorad) following the manufacturer's recommendations. All data are presented as relative expression units after normalization to the HPRT gene.

## Statistics and reproducibility

Data are presented as mean ± SEM. Student's *t* test, one-way or two-way analysis of variance (ANOVA) followed by the Tukey or Dunnett's post hoc test was applied for comparison between two groups or among multiple groups using Graphpad Prism 10.0, respectively. Non-parametric data were analyzed using Kruskal–Wallis's test and Dunn's multiple comparison post-test. Additionally, the Mantel–Cox test and Two-tailed, unpaired Spearman correlation was utilized for conducting survival and weight analysis, respectively. $p < 0.05$ was considered to be statistically significant. Two-tailed, unpaired Spearman correlation. Each experiment is repeated at least three times independently with similar results and the representative dataset is presented.

## Reporting summary

Further information on research design is available in the Nature Portfolio Reporting Summary linked to this article.

## Data availability

The data that support the findings of this study are available within the paper and its Supplementary Information or from the corresponding author upon reasonable request. Source data are provided with this paper.

## Code availability

All code generated in this study are available from the corresponding author upon on reasonable request.

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

## Acknowledgements

This work was funded in part by National Council for Scientific and Technological Development-CNPq (401390/2020-9; 442731/2020-5; 305932/2022-5; 422002/2023-2; 408482/2022-2; 465425/2014-3), PRPq-UFMG, CAPES (88887.506690/2020-00; 38/2022 Programa de desenvolvimento da pós-graduação parcerias estratégicas nos estados III), FAPEMIG (APQ-00826-21; APQ-02402-23; RED-00202-22 Rede de Pesquisa em Imunobiológicos e Biofármacos para terapias avançadas e inovadoras); MCTI/FINEP—MS/SCTIE/DGITIS/CGITS (6205283B-BB28-4F9C-AA65-808FE4450542); INCT em Dengue e Interação Microorganismo-hospedeiro (MCTI/CNPq/CAPES/FAPs 16/2014). P.P.G.G. is supported by CNPq (442731/2020-5; 305932/2022-5; 422002/2023-2; 408482/2022-2); FAPEMIG (APQ-00826-21; APQ-02402-23); and MCTI/FINEP—MS/SCTIE/DGITIS/CGITS (6205283B-BB28-4F9C-AA65-808FE4450542).

## Author contributions

P.P.G.G., F.F., V.V.C., M.M.T., L.C.G., P.A.C.C. designed the experiments. L.C.G., P.A.C.C., H.A.S.F., S.R.A.S.J., L.C.O., A.C.S.B., M.M.F., C.M.Q.J., W.N.S., N.J.A.S., A.K.S., K.K.S.F., M.T.R.A. and F.M.M., performed experiments and collected data. P.P.G.G., F.F., V.V.C., M.M.T., L.C.G., P.A.C.C., H. F., A. B., A. T. C., R. S. A., M.J. M., S. S., A. H. discussed the results and strategy. P.P.G.G., L.C.G., P.A.C.C., S.R.A.S.J. wrote the manuscript, which was edited by all co-authors. P.P.G.G., F.F. supervised, directed and managed the study.

## Competing interests

L.C.G., A.C.S.B., H.A.S.F., F.F. and P.P.G.G. are inventors on a patent filed by the Federal University of Minas Gerais (BR102023011998, 2023) describing the lipid nanoparticle formulation technology in this manuscript. All remaing authors declare no competing interests.

## Additional information

Lays Cordeiro Guimaraes [1], Pedro Augusto Carvalho Costa [1], Sérgio Ricardo Aluotto Scalzo Júnior [1], Heloísa Athaydes Seabra Ferreira [1], Ana Carolina Soares Braga [1], Leonardo Camilo de Oliveira[2], Maria Marta Figueiredo[3], Sarah Shepherd[4], Alex Hamilton [4], Celso Martins Queiroz-Junior[5], Walison Nunes da Silva [1], Natália Jordana Alves da Silva [1], Marco Túllio Rodrigues Alves[1], Anderson Kenedy Santos[1], Kevin Kelton Santos de Faria [1], Fernanda Martins Marim[6], Heidge Fukumasu[7], Alexander Birbrair [8], Andréa Teixeira-Carvalho[9], Renato Santana de Aguiar [6], Michael J. Mitchell [4], Mauro Martins Teixeira[2], Vivian Vasconcelos Costa[5], Frederic Frezard [1] & Pedro Pires Goulart Guimaraes [1] ✉

[1]Department of Physiology and Biophysics, Institute of Biological Sciences, Federal University of Minas Gerais, Belo Horizonte 31270-901 Minas Gerais, Brazil. [2]Department of Biochemistry and Immunology, Federal University of Minas Gerais, Belo Horizonte 31270-901 Minas Gerais, Brazil. [3]State University of Minas Gerais, Divinopolis 35501-170 Minas Gerais, Brazil. [4]Department of Bioengineering, University of Pennsylvania, Philadelphia 19104 PA, USA. [5]Department of Morphology, Federal University of Minas Gerais, Belo Horizonte 31270-901 Minas Gerais, Brazil. [6]Department of Genetics, Federal University of Minas Gerais, Belo Horizonte 31270-901 Minas Gerais, Brazil. [7]Department of Animal Science and Food Engineering, University of São Paulo, Pirassununga 13635-900 São Paulo, Brazil. [8]Department of Dermatology, University of Wisconsin-Madison, Madison 53706 WI, USA. [9]Grupo Integrado de Pesquisas em Biomarcadores, Instituto René Rachou, Fundação Oswaldo Cruz, Belo Horizonte 30190-009 Minas Gerais, Brazil. ✉e-mail: ppiresgo@reitoria.ufmg.br

