## [Peer Review File · Nature Communications]

Nanoparticle-based DNA vaccine protects against SARS-CoV-2 variants in female preclinical modelsEditorial Note: Some elements on pages 18, 24, 25, 41 & 42 have been reproduced from Servier Medical Art under a [CC BY 3.0 Attribution](https://creativecommons.org/licenses/by/3.0/) licence. Some elements on pages 14, 15, 27, 34 & 46 have been redacted as indicated to remove third-party material where no permission to publish could be obtained.

REVIEWER COMMENTS

Reviewer #1 (Remarks to the Author):

Review of "Nanoparticle-based DNA vaccine induced protective effect against SARS-CoV2 2 variants in preclinical model" by Guimaraes et al.

Overview. Guimaraes et al ask an important question in the field of nucleic acid vaccines, can DNA vaccines be successfully formulated with lipid nanoparticles (LNP). LNPs were an essential component of the successful mRNA vaccines approved for use against COVID-19. In contrast, DNA vaccines were delivered by other means such as jet (ZyCoV-D) or in vivo electroporation (INO-4800), which may not be ideal due to the need for a delivery device. Guimaraes and colleagues provide data concerning the formulation of a DNA vaccine using a cocktail of ingredients and microfluidics. Identifying the correct molar ratios of ingredients appears important in successful formulation of the LNP-DNA complexes. The authors then proceed to demonstrate biodistribution of their barcoded DNA and in vitro uptake by immune cells, and identify the optimal formulation. Using one of these LNPs they formulate a COVID-19 vaccine using the HexPro spike construct. Authors should define in the main text whether this is matched to WT or another SARS-CoV-2 VOC spike. The team proceeds to assess the immunogenicity and efficacy of this construct in the mouse model. The article is very well written and easy to follow. However, I have major concerns which need to be addressed.

Major Concerns

While the step by step process to identify and test an optimal LNP formulation for DNA is impressive the following immune analysis of the LNP formulated COVID19 DNA vaccine is weak.

1. The only comparator tested appears to be an empty DNA plasmid formulated with the LNP. I would strongly suggest a non-LNP formulated ie naked HexaPro DNA vaccine group should have been run head to head in this analysis. Additionally, a mRNA LNP comparator should have also been used, for example one of the approved COVID-19 vaccines (Moderna or Biontech/Pfizer). LNP formulations may be successful in protecting the nucleic vaccine from enzyme degradation and assisting in delivery into the target cell cytoplasm, but DNA vaccines require entry into the cell nucleus. This requirement may make such formulations not optimal for DNA vaccines. This leads to point 2. The team use a very high dose of DNA (45 ug) in the mouse studies and fail show complete protection in the challenge study. Other groups have demonstrated protection in preclinical models with just naked DNA, and the LNP-mRNA vaccines showed high levels of protection at lower doses. Without benchmarking the described vaccine against the two comparators described above, the reader cannot draw any meaningful conclusions on the DNA-LNP vaccine's performance either at the level of immunogenicity or efficacy.

3. Additionally, the evaluation of the vaccine has only been performed in a single species. A major challenge with DNA vaccines has been scaling up. Thus, at least the immunogenicity of the LNP-DNA vaccine should have been assessed in a larger animal model and against comparators. LNP DNA vaccines could offer very attractive alternatives to mRNA-LNP vaccines, but answering the above questions is required.

Reviewer #2 (Remarks to the Author):

In this manuscript, the authors described and screened new LNP formulations for a DNA-based SARS-CoV-2 vaccine and showed that the selected formulation is immunogenic and provided partial protection against SARS-CoV-2 VOC Gamma lineage (P.1) in a lethal mouse model. Formulations of nucleic acid-based vaccines is critical for safety, immunogenicity and associated protection of vaccines. Therefore, this work brings important contribution to the vaccine field.

However, to be considered for publication the manuscript needs substantial improvement. Especially critical is providing antibody data pre-challenge (point 4).

Comments:

1) In the abstract, please mention which animal models were used and sex, according to 'Sex and Gender Equity in Research – SAGER – guidelines'.

2) In the introduction, in the last paragraph it is indicated that in vitro and in vivo efficacy and immunogenicity of the lead LNP formulated with recombinant HexaPro spike plasmid DNA against SARS-CoV-2 VOC Gamma lineage (P.1) and Omicron (B.1.1.529) were assessed. However, the efficacy was only evaluated against SARS-CoV-2 VOC Gamma lineage (P.1).

- Please specify this, as now the statement does not fully reflect the data.

- Please also clarify it in the abstract.

- Can you please include in the manuscript the rationale for testing efficacy against SARS-CoV-2 VOC Gamma lineage (P.1), considering that the most relevant variant at this moment is Omicron?

3) Line 133: b-DNA-LNPs to target lymphoid tissues.

To evaluate the biodistribution of 15 LNP formulations encapsulating unique DNA barcodes (b-DNA-LNPs), the b-DNA-LNPs were pooled together and administered via intramuscular injection into mice. After evaluating the biodistribution of 15 formulations, the two best LNPs were selected.

- For the immune response an important compartment to look is represented by the draining lymph nodes (i.e. iliac and inguinal lymph nodes), as they are the location of the primary germinal center and therefore antibody response. This is a gap that needs to be addressed in the manuscript.

- Can the authors explain how it is controlled that the delivery of each b-DNA-LNP it is not influenced by the presence of the other 14 b-DNA-LNPs?

- And therefore, how is it controlled that promising b-DNA-LNPs are not discarded?

- How homogeneous are the different formulations?

- Were the different formulations tested individually in-vitro for transfection efficiency, as described for the best two LNPs, B4 and B10?

4) The antibody responses are shown at 5 days post-infection (Figure 6). The virus provides an additional dose of antigen and therefore boosts immune responses. To evaluate the immune responses solely elicited by the vaccine:

- The authors should include spike-specific IgG levels and SARS-CoV-2 antibody neutralizing titers pre-challenge. This should be possible as according to the methods, blood samples were collected at day 36 (pre-challenge). Can these data be added to the manuscript?

- To better appreciate the variability among the different animals, can the author provide the data in the graphs per single animal?

- In lines 234-237 it is stated: We next evaluated the levels of neutralizing antibodies in the sera of immunized mice against SARS-CoV-2 variants of Gamma (P.1) and Omicron (B.1.1.529) lineage. We observed that LNP-HPS induced neutralizing antibodies that protected against SARS-CoV-2 variants of Gamma (P.1) and Omicron (B.1.1.529) lineage (Fig. 6D-E). The word protects can be confused with in-vivo protection. Please consider changing it.

5) The expression levels of antiviral genes and cytokines in lungs immunized mice, were measured by quantitative PCR. The authors indicated that there was enhanced expression levels of TNF and IFN- γ in mice immunized with LNP-HPS compared to control groups and that for the other investigated cytokines there were no significant differences between groups.

- Based on Fig. S5 significance is only reported for TNF against mock control. The most relevant comparisons are against PBS and LNP-C. The comparisons showing significant differences should be described in the results.

- Is statistic missing in the graph for IFN- γ ? In case there is no significant difference for IFN- γ among the different groups, the statement that expression levels are enhanced compared to other groups is not correct and needs to be adjusted.

- Similarly, also IL-6 gene expression seems higher in mice immunized with LNP-HPS. Is statistic missing in the graph? IL-6 is a pro-inflammatory cytokine associated with severe disease in COVID-19. This observation may deserve more attention as could flag a safety signal. Consider discussing it.

6) In the discussion, lines 319-321 it is stated: Also, the remarkable resistance against variant P.1 was confirmed by the reduced, if not absent, viral load in the lung of mice immunized with LNP-HPS, 320 as determined via RT-qPCR, plaque assays, and immunofluorescence (Fig. 4E-I).

- This is not fully reflecting the data as by RT-qPCR (Fig 4G) viral load is clearly detected, but

significantly lower compared with the LNP-C group. Please correct the statement. In Fig 4G, the comparison with PBS is missing and looks as the one with LNP-C. For consistency with the other panels, please add it for panel G.

- For immunofluorescence also the mock group is included. Is there a specific reason for it? Why is it not included for the other readouts?

7) In the discussion, lines 325-328 it is stated: Intramuscular vaccination with LNP-HPS induced very high titers of antigen-specific IgG in sera as well as enhanced IgA titers in lung homogenate (Fig. 6B-C), suggesting the role of IgA in the early stage of neutralizing response to SARS-CoV-2.

- As indicated in point 4), this statement can only be supported by showing antibody data (spike-specific IgG and SARS-CoV-2 antibody neutralizing titers) pre-challenge. I hope these data can be added.

8) In the methods, please include:

- The sex of the K18-hACE2 mice, now only age is indicated.

- Where the animal experiments (in C57BL/6 and K18-hACE2 mice) were performed and the committees that approved it.

9) Nowhere is mentioned on which SARS-CoV-2 lineage is based the spike of the vaccine. Can this be indicated in the manuscript?

Reviewer #3 (Remarks to the Author):

In "Nanoparticle-based DNA vaccine induced protective effect against SARS-CoV2 2 variants in preclinical model", Prof. Guimaraes and collaborators developed lipid nanoparticles delivering barcoded-DNA (b-DNA) as vaccines for COVID19. The authors used bDNA encoding the proteins of SARS-CoV2 variants, which current vaccine formulations are partially effective. Lipid nanoparticles have demonstrated high performance for RNA and DNA vaccines against COVID-19. In fact, a COVID-19 DNA vaccine based on lipid nanoparticles has been clinically approved, and various formulations are under intense clinical investigation [The Lancet 399, 1313–1321 (2022); The Lancet 399, 1281–1282 (2022); Nature 597, 161–162 (2021); etc.]. Current study extends the application of lipid nanoparticles to make DNA vaccines against elusive SARS-CoV2 variants. To identify the formulations, the authors used high-throughput screening, which is common in the development of lipid-based vaccines. Thus, while the findings are interesting, I think the work is merely augmenting the existing knowledge rather than introducing any groundbreaking discoveries. The study's methodology is rigorous and adheres to the typical standards in the field. Nevertheless, the conclusions and assertions made are only partially substantiated by the data, and supplementary evidence is required to further support them. Therefore, I would like to note the following points:

1. In the Abstract: "LNP-HPS elicited a robust protective effect against VOCs, as evidenced by improved clinical outcomes and significantly reduced lethality...". The word "clinical" may be a misspelling. I guess the authors are referring to "pre-clinical"

2. In the Introduction section there are several points that the authors should reconsider, as follows:

a. In line 93 the authors indicated "In addition, DNA is cost-effective, easy to produce, more stable than mRNA, and does not require frozen transport, which can be relevant for remote areas and developing nations". Actually, the problem of frozen transport has been recently solved in new LNP formulations, which can be transported at higher temperatures (<https://www.nature.com/articles/s41421-022-00517-9>).

b. The authors mentioned in line 96 "Whereas there is one DNA vaccine approved for COVID-19 (ZyCoV-D) and at least 10 in clinical trials (Phase I-III), such as INO-4800, AG0301-COVID19, and GX98 19N, vaccine strategies using DNA-loaded nanoparticles remain in pre-clinical studies". These numbers, as well as DNA vaccines for other diseases, such as cancer, actually show DNA vaccine technology is already mature. The authors should reconsider this statement.

3. The dynamic light scattering (DLS) technique used in this study does not provide results below 1 nm. Thus, showing DLS results with higher resolution than 1 nm is incorrect. The authors should correct all the diameters.

4. The biodistribution studies lack critical tissues, such as muscle, draining lymph nodes, distant

lymph nodes, etc. How is the distribution to these tissues?

5. In vitro studies evaluated the expression in macrophages, fibroblasts and endothelial cells. The authors should study more relevant cells to the administration route and the immune response, such as dendritic cells, muscle cells, B cells, etc.

6. Besides the cytotoxicity, how is the immune response to the lipids making the vaccine?

7. The safety is not demonstrated. The toxicity of the LNPs against the different tissues (muscle, liver, spleen, heart, etc.) should be confirmed.

8. The post-vaccination serum cytokines levels should be determined. Is there a correlation with the cytokines with the antibody titers and the vaccine protection? By examining these relationships, the significance of the study can be improved.

9. Could the efficacy of the vaccines be tested against different strains of SARS-CoV2 to determine their suitability? This additional experiment could enhance the significance of the research.

REVIEWER COMMENTS AND AUTHORS' ANSWERS

We thank the reviewers for their insightful comments and their time to help us improve our manuscript. We have revised the manuscript in accordance with the comments received, which we believe have significantly improved our manuscript.

Specifically, we executed experiments to corroborate the *in vitro* data, *in vivo* biodistribution, immunogenicity and efficacy of our nanoparticle-based DNA vaccine (LNP-HPS). As suggested by Reviewer #1, we compared the immunogenicity and efficacy of our LNP-HPS to naked DNA and an approved COVID-19 vaccine from Biontech/Pfizer (BNT-mRNA) as well as all the other previously mentioned groups (Mock, PBS, and control LNP) in mice. Further, we assessed the immunogenicity of our LNP-HPS in larger animals (hamsters) compared to all the reference groups (Mock, PBS, naked DNA, control LNP, and BNT-mRNA). Additionally, we also evaluated the homogeneity and biodistribution of 15 LNP formulations encapsulating unique DNA barcodes (b-DNA-LNPs) in draining lymph nodes and muscle tissue as suggested by reviewers #2 and #3. In response to the suggestion from Reviewer #2, we also executed experiments to assess spike-specific IgG levels and SARS-CoV-2 antibody neutralizing titers pre-challenge in both mice and hamsters, providing data in the graphs per single animal (for pre and post-challenge). As suggested by Reviewer #3, we executed additional *in vitro* studies to evaluate the expression in dendritic cells, muscle cells and B cells induced by the top-performers LNPs (now we have data in macrophages, fibroblasts, endothelial cells, dendritic cells, muscle cells and B cells). In addition, addressing the suggestion from Reviewer #3, we evaluated the toxicity of our LNPs against the different tissues of mice and hamsters (muscle, liver, spleen, heart) and the post-vaccination serum cytokines levels were determined. Lastly, we assessed the SARS-CoV-2 antibody neutralizing titers and specific IgG levels both pre-challenge and post-challenge against SARS-CoV-2 variants of concern (VOCs). Collectively, our revised manuscript provided favorable biodistribution, including in lymph nodes and muscle, depth characterization of the immune response post vaccination (pre and post-challenge) in mice and larger animals. The findings indicate superior immunogenicity and protection efficacy conferred by our LNP-HPS in comparison to naked DNA, and comparable immunogenicity and protection efficacy to the approved COVID-19 vaccine from Biontech/Pfizer (BNT-mRNA). Finally, additional methodological details have been added accordingly with the sole goal to improve our manuscript for the general readership and we have directly addressed all the points raised by the reviewers.

All the consequential changes are highlighted in yellow in our resubmission. Specific discussion of the reviewers' concerns is present in our point-by-point response below.

REVIEWER COMMENTS

Reviewer #1 (Remarks to the Author):

Review of "Nanoparticle-based DNA vaccine induced protective effect against SARS-CoV2 2 variants in preclinical model" by Guimaraes et al.

Overview. Guimaraes et al ask an important question in the field of nucleic acid vaccines, can DNA vaccines be successfully formulated with lipid nanoparticles (LNP). LNPs were an essential component of the successful mRNA vaccines approved for use against COVID-19. In contrast, DNA vaccines were delivered by other means such as jet (ZyCoV-D) or *in vivo* electroporation (INO-4800), which may not be ideal due to the need for a delivery device. Guimaraes and colleagues provide data concerning the formulation of a DNA vaccine using a cocktail of ingredients and microfluidics. Identifying the correct molar ratios of ingredients appears important in successful formulation of the LNP-DNA

complexes. The authors then proceed to demonstrate biodistribution of their barcoded DNA and in vitro uptake by immune cells, and identify the optimal formulation. Using one of these LNPs they formulate a COVID-19 vaccine using the HexPro spike construct. Authors should define in the main text whether this is matched to WT or another SARS-CoV-2 VOC spike. The team proceeds to assess the immunogenicity and efficacy of this construct in the mouse model. The article is very well written and easy to follow. However, I have major concerns which need to be addressed.

Response: We thank the reviewer for their remarkable feedback on our manuscript, as well as for their time and effort in helping us improve it. We have defined the HexaPro spike construct in the main text, as suggested.

Results: ...“Hexapro is a recombinant spike protein, derived from the SARS-CoV-2 wild-type, with additional six-proline substitutions (F817P, A892P, A899P, A942P), which exhibited higher expression than two-proline substitutions²⁶. The two-proline substitution in the S2 subunit was used in the mRNA sequences of RNA vaccines approved²⁷“...

Major Concerns

While the step by step process to identify and test an optimal LNP formulation for DNA is impressive the following immune analysis of the LNP formulated COVID19 DNA vaccine is weak.

1. The only comparator tested appears to be an empty DNA plasmid formulated with the LNP. I would strongly suggest a non-LNP formulated ie naked HexaPro DNA vaccine group should have been run head to head in this analysis. Additionally, a mRNA LNP comparator should have also been used, for example one of the approved COVID-19 vaccines (Moderna or Biontech/Pfizer). LNP formulations may be successful in protecting the nucleic vaccine from enzyme degradation and assisting in delivery into the target cell cytoplasm, but DNA vaccines require entry into the cell nucleus. This requirement may make such formulations not optimal for DNA vaccines. This leads to point. The team use a very high dose of DNA (45 ug) in the mouse studies and fail show complete protection in the challenge study. Other groups have demonstrated protection in preclinical models with just naked DNA, and the LNP-mRNA vaccines showed high levels of protection at lower doses. Without benchmarking the described vaccine against the two comparators described above, the reader cannot draw any meaningful conclusions on the DNA-LNP vaccine's performance either at the level of immunogenicity or efficacy.

Response: We thank the reviewer for the excellent feedback. To address this comment, we assessed the immunogenicity and efficacy of our LNP-HPS compared to naked DNA and approved COVID-19 vaccine from Biontech/Pfizer (BNT-mRNA). Consequently, we expanded the experiment to include all the previously mentioned groups (Mock, PBS, and control LNP), as well as naked DNA and BNT-mRNA, in both mice and larger animals (hamsters). Our findings demonstrate superior immunogenicity and protection efficacy conferred by our LNP-HPS compared to naked DNA. Furthermore, we observed comparable immunogenicity and protection of LNP-HPS to the approved COVID-19 vaccine from Biontech/Pfizer (BNT-mRNA) in mice. Additionally, we appreciate the reviewer's attention to detail regarding the typo in the DNA dose, and we sincerely apologize for any confusion. The correct information is as follows: Mice and hamsters received two doses, corresponding to 22.5 µg and 67.5 µg of DNA per dose, respectively, administered at a 21-day interval. The total dose (prime and boost) was 45 µg and 135 µg for mice and hamsters, respectively.

Fig. 4: LNP-based DNA vaccine protects K18-ACE-2 mice against SARS-CoV-2 variants Gamma lineage (P.1) (A) Scheme of immunization: K18-ACE-2 mice were immunized with controls and LNP-HPS and boosted with equivalent doses 3 weeks later. Immunized mice were challenged with lethal 6×10^4 PFU of SARS-CoV-2 variants Gamma lineage (P.1) 15 days after boost. (B, C) The lethality and weight loss were

monitored for 15 days after challenge (n = 7/group). **(D)** Scheme of immunization wherein the lungs were harvested 5 days post-infection for all immunized groups. **(E, F)** The plaque-forming unit measures the viable Vero cells treated with serum of immunized mice 5 days post-infection (n = 5/group). Dashed line denotes limit of detection. **(G)** SARS-CoV-2 genome copies detected in the lung of the immunized mice at 5 days post-infection (n = 4-5/group). **(H)** Analysis of infection via spike protein in lung sections of immunized mice at 5 days post-infection via immunofluorescence from at least 20 fields of 3 different sections (n = 5/group). **(I)** Representative fluorescence images of lungs marked with anti-Spike antibody. Samples were stained with anti-SARS-CoV-2 Spike protein (RBD) antibody (magenta) and DAPI (blue) and acquired using a 20x objective (n = 5/group). Data are presented as mean \pm SEM; *p<0.05, **p<0.01, ***p<0.001, ****p<0.0001. **(B)** Log-rank (Mantel–Cox) test. **(C)** Two-tailed, unpaired Spearman correlation to test. **(E, H)** One-way ANOVA with Dunnet's post hoc test. **(G)** Kruskal–Wallis followed by Dunn's multiple comparisons test.

Fig. 5: LNP-HPS vaccine reduced lung damage. (A) Scheme of immunization. K18-ACE-2 mice were immunized with controls and LNP-HPS and boosted with equivalent doses 3 weeks later. Immunized mice were challenged with lethal 6×10^4 PFU of SARS-CoV-2 variants Gamma lineage (P.1). Lungs were harvested at 5 days post-infection for all immunized groups for histopathological analysis. **(B)** Histopathological analysis at 20 x magnification of the lungs at 5 days post-infection with the P.1 strain (n = 5/group).

Histopathological score for (C) pleura, (D) bronchi, (E) alveoli and (F) vessels ($n = 5/\text{group}$). The arrows indicate pleura (pink), bronchi (blue), alveoli (red), edema (yellow), vessels (green) and vessels congestion (orange). Data are presented as mean \pm SEM; * $p < 0.05$, ** $p < 0.01$, *** $p < 0.001$. Two-way ANOVA followed by Tukey's multiple comparison test.

Fig. 6: LNP-based DNA vaccine induced potent immune response and protection against SARS-CoV-2 infection and its variants (A) Scheme of immunization. K18-ACE-2 mice were immunized with controls and LNP-HPS, and boosted with equivalent doses 3 weeks later. Immunized mice were challenged with lethal 6×10^4 PFU of SARS-CoV-2 variants Gamma lineage (P.1), sera were collected at 24 h before the infection (pre-infection), and 5 days post-infection for all immunized groups and the lungs were collected 5 days post-infection. Levels of total antigen-specific IgG anti-spike serum of mice at (B) pre-infection, and (C) 5 days post-infection ($n = 5/\text{group}$). (D) Levels of total IgA anti-spike in lung homogenate at 5 days post-infection ($n = 5/\text{group}$). Levels of neutralizing antibodies in sera of immunized mice at (E, F) pre-infection, and (G, H) 5 days post-infection against P.1 ($n = 5/\text{group}$). Levels of neutralizing antibodies in sera of immunized mice at (I, J) pre-infection, and (K, L) 5 days post-infection against Omicron ($n = 5/\text{group}$). Data are presented as mean \pm SEM; ** $p < 0.01$. (F, H, J, L) Kruskal-Wallis followed by Dunn's multiple comparisons test.

Fig. 7: LNP-based DNA vaccine-induced antigen-specific T cell response. (A) Scheme of immunization. K18-ACE-2 mice were immunized with controls and LNP-HPS and boosted with equivalent doses 3 weeks later. Immunized mice were challenged with lethal 6×10^4 PFU of SARS-CoV-2 variants Gamma lineage (P.1). Spleens were harvested 5 days post-infection for all immunized groups and splenocytes were stimulated with RBD-S1 protein. The graph shows cell number of **(B)** CD8⁺ and **(E)** CD4⁺

T cells producers of **(C, F)** IFN- γ and **(D, G)** Gran-b ($n=4-5$ samples/group). Cell number of central memory and effector/effector memory from **(H, N)** CD8⁺ and **(K, Q)** CD4⁺ T cells; and **(I, L, O, R)** IFN- γ and **(J, M, P, S)** Gran-b production from memory subsets ($n=4-5$ samples/group). Data are presented as mean \pm SEM; * $p<0.05$, ** $p<0.01$, *** $p<0.001$, **** $p<0.0001$. One-way ANOVA followed by Tukey's multiple comparison test.

3. Additionally, the evaluation of the vaccine has only been performed in a single species. A major challenge with DNA vaccines has been scaling up. Thus, at least the immunogenicity of the LNP-DNA vaccine should have been assessed in a larger animal model and against comparators.

LNP DNA vaccines could offer very attractive alternatives to mRNA-LNP vaccines, but answering the above questions is required.

Response: We thank the reviewer for the thoughtful comment. We have performed the experiments in larger animals (hamsters), incorporating the recommended groups: naked DNA and approved COVID-19 vaccine from Biontech/Pfizer (BNT-mRNA). Our findings substantiate the superior immunogenicity of our LNP-HPS in comparison to naked DNA, and we observe comparable immunogenicity to the approved COVID-19 vaccine from Biontech/Pfizer (BNT-mRNA) in hamsters, aligning with the reviewer's suggestion.

Fig. 8: Immunogenicity and efficacy of LNP-based DNA vaccine against SARS-CoV-2 variants Gamma lineage (P.1) in hamsters. (A) Scheme of immunization: Syrian hamsters were immunized with controls and LNP-HPS and boosted with equivalent doses 3 weeks later. Immunized hamsters were challenged with 6×10^5 PFU of SARS-CoV-2 variants Gamma lineage (P.1) 15 days after boost. Lungs were harvested 5 days post-infection for all immunized groups. **(B)** The weight loss was

monitored for 5 days after challenge (n = 5/group). **(C)** SARS-CoV-2 genome copies detected in the lung of the immunized hamsters at 5 days post-infection (n = 4-5/group). **(D, E)** The plaque-forming unit measures the viable Vero cells treated with serum of immunized hamsters 5 days post-infection (n = 5/group). Dashed line indicates limit of detection. **(F)** Histopathological analysis at 20 x magnification of the lungs at 5 days post-infection with the P.1 strain (n = 5/group). Histopathological score for **(G)** pleura, **(H)** bronchi, **(I)** alveoli and **(J)** vessels (n = 5/group). The arrows indicate pleura (pink), bronchi (blue), alveoli (red), edema (yellow), vessels (green) and vessels congestion (orange). Data are presented as mean \pm SEM; *p<0.05, **p<0.01, ***p<0.001. **(B)** Two-tailed, unpaired Spearman correlation to test. **(C)** Kruskal–Wallis followed by Dunn’s multiple comparisons test. **(D)** One-way ANOVA with Dunnet’s post hoc test. **(G–J)** Two-way ANOVA followed by Tukey’s multiple comparison test.

Fig. 9: LNP-based DNA vaccine induced potent immune response against SARS-CoV-2 infection and its variants in hamsters. (A) Scheme of immunization. Syrian hamsters were immunized with controls and LNP-HPS, and boosted with equivalent doses 3 weeks later. Immunized mice were challenged with 6×10^5 PFU of SARS-CoV-2 variants Gamma lineage (P.1), sera was collected at 24 h before the infection and 5 days post-infection for all immunized groups. Levels of total antigen-specific IgG anti-spike serum of hamsters **(B)** at pre-infection, and **(C)** 5 days post-infection (n = 5/group). Levels of neutralizing antibodies in sera of immunized hamsters at **(D, E)** pre-infection, and **(F, G)** 5 days post-infection against P.1 (n = 5/group). Levels of neutralizing antibodies in sera of immunized hamsters at **(H, I)** pre-infection, and **(J, K)** 5 days post-infection against Omicron (n = 5/group). Data are presented as mean \pm SEM; **p<0.01. (E,I,G,K) Kruskal-Wallis followed by Dunn’s multiple comparisons test.

Reviewer #2 (Remarks to the Author):

In this manuscript, the authors described and screened new LNP formulations for a DNA-based SARS-CoV-2 vaccine and showed that the selected formulation is immunogenic and provided partial protection against SARS-CoV-2 VOC Gamma lineage (P.1) in a lethal mouse model.

Formulations of nucleic acid-based vaccines is critical for safety, immunogenicity and associated protection of vaccines. Therefore, this work brings important contribution to the vaccine field.

However, to be considered for publication the manuscript needs substantial improvement. Especially critical is providing antibody data pre-challenge (point 4).

Response: We thank the reviewer for their remarkable feedback on our manuscript, as well as for their time and effort in helping us improve it.

Comments:

1) In the abstract, please mention which animal models were used and sex, according to 'Sex and Gender Equity in Research – SAGER – guidelines'.

Response: We thank the reviewer for the comment. We have addressed the comment appropriately. Please see changes in the abstract as suggested.

Abstract: ...“Next, the lead LNP was used to encapsulate HexaPro plasmid (LNP-HPS) to evaluate protective efficacy against SARS-CoV-2 VOC Gamma lineage (P.1) and the immunogenicity against both P.1 and and Omicron (B.1.1.529) VOCs **in female mice and hamsters**”...

Title: Nanoparticle-based DNA vaccine induced protective effect against SARS-CoV-2 variants **in female** preclinical models

2) In the introduction, in the last paragraph it is indicated that in vitro and in vivo efficacy and immunogenicity of the lead LNP formulated with recombinant HexaPro spike plasmid DNA against SARS-CoV-2 VOC Gamma lineage (P.1) and Omicron (B.1.1.529) were assessed. However, the efficacy was only evaluated against SARS-CoV-2 VOC Gamma lineage (P.1).

- Please specify this, as now the statement does not fully reflect the data.

Response: We thank the reviewer for bringing this up to our attention. We have addressed the comment appropriately. Please, see the changes as suggested:

Introduction: ...” **Next, we assessed the protective efficacy of our leading LNP, namely LNP-HPS, formulated with recombinant HexaPro spike plasmid DNA against SARS-CoV-2 Gamma lineage (P.1).** Additionally, we evaluated the **immunogenicity against SARS-CoV-2 VOCs P.1 and Omicron (B.1.1.529)** (Fig. 1B). Our findings reveal that a two-dose immunization of our lead LNP-based DNA vaccine elicited a robust protective effect against P.1 and induced production of IFN- γ and granzyme-B response by T cells in K18-ACE-2 mice. Importantly, it also led to a reduced viral load in the lungs and enhanced antigen-specific IgG titers in both mice and hamsters. Furthermore, LNP-HPS generated high levels of neutralizing antibodies against SARS-CoV-2 VOCs P.1 and Omicron in both animal models. Together, our data demonstrate the versatility of our nanoparticle-based DNA vaccine platform, suggesting its potential utility in preventing SARS-CoV-2 infection”...

- Please also clarify it in the abstract.

Response: We thank the reviewer for bringing this up to our attention. We have addressed the comment appropriately. Please see changes in the abstract as suggested:

Abstract: ...” LNP-HPS elicited a robust protective effect against P.1, **correlating with reduced lethality, decreased viral load in the lungs and reduced lung damage. Importantly, LNP-HPS induced potent humoral and T cell responses against P.1, and generated high levels of neutralizing antibodies against P.1 and Omicron. Our findings indicate that the protective efficacy and immunogenicity elicited by LNP-HPS were comparable to those achieved by the approved COVID-19 vaccine from Biontech/Pfizer in both animal models. Together, these findings suggest that LNP-HPS holds great promise as a vaccine candidate against VOCs.**”...

- Can you please include in the manuscript the rationale for testing efficacy against SARS-CoV-2 VOC Gamma lineage (P.1), considering that the most relevant variant at this moment is Omicron?

Response: We thank the reviewer for the excellent point. We have incorporated the rationale for testing efficacy against SARS-CoV-2 VOC Gamma lineage (P.1) in the manuscript.

Discussion: While Omicron currently holds significance as a VOC, this variant induces milder disease in mice and hamsters^{37,38}. In contrast, the Gamma lineage (P.1) has exhibited heightened virulence, pathogenicity, and lethality in K18-hACE2 mice compared to other variants³⁹. Consequently, we opted for the Gamma lineage (P.1) to evaluate the protective efficacy of LNP-HPS vaccination. Additionally, we examined the neutralizing titers against Omicron using sera from immunized mice both before and after infection.

References:

³⁷Halfmann, P. J. et al. SARS-CoV-2 Omicron virus causes attenuated disease in mice and hamsters. *Nature* 603, 687–692 (2022).

³⁸Suzuki, R. et al. Attenuated fusogenicity and pathogenicity of SARS-CoV-2 Omicron variant. *Nature* 603, 700–705 (2022).

³⁹Stolp, B. et al. SARS-CoV-2 variants of concern display enhanced intrinsic pathogenic properties and expanded organ tropism in mouse models. *Cell Rep* 38, 110387 (2022).

3) Line 133: b-DNA-LNPs to target lymphoid tissues.

To evaluate the biodistribution of 15 LNP formulations encapsulating unique DNA barcodes (b-DNA-LNPs), the b-DNA-LNPs were pooled together and administered via intramuscular injection into mice. After evaluating the biodistribution of 15 formulations, the two best LNPs were selected.

- For the immune response an important compartment to look is represented by the draining lymph nodes (i.e. iliac and inguinal lymph nodes), as they are the location of the primary germinal center and therefore antibody response. This is a gap that needs to be addressed in the manuscript.

Response: We thank the reviewer for the thoughtful comment, and we agree. In response to this comment, we evaluated the biodistribution of the 15 LNP formulations encapsulating unique DNA barcodes (b-DNA-LNPs) in draining lymph nodes and muscle. The results have been incorporated into Figure 2 and Supplementary Figure 1, as well as the main manuscript."

Fig. 2: Top-performing LNPs induced enhanced in vitro GFP expression in different cell lines. (A) Schematic of the 2 top-performing LNPs encapsulating DNA encoding GFP for in vitro screening of transfection efficiency in four different cell lines. **(B, F, J, N)** Quantification of GFP expression was measured after 24 h, 48 h and 72 h, and cell viability was measured after 72 h in **(D)** macrophages (RAW 264.7), **(H)** dendritic cells (JAWSII), and **(L)** fibroblasts (L929), and **(M)** myoblasts (C2C12) transfected with LNP B4 at different doses ($n = 3/\text{group}$). **(C, G, K, O)** Representative GFP fluorescence (Bottom) and overlaid on brightfield (Top) photomicrographs after treatment of LNP B4

in four different cell lines ($n = 4/\text{group}$). **(E, I, M, Q)** Quantification of GFP expression after treatment with B4 and B10 LNPs in different cell lines. Data are presented as mean \pm SEM ($n = 4/\text{group}$). Data are presented as mean \pm SEM. **(B, F, J, N D, H, L, P)** One-way ANOVA with Dunnet's post hoc test; * $p < 0.05$, *** $p < 0.001$ **** $p < 0.0001$. **(E, I, M, Q)** Two-tailed unpaired t-test; ** $p < 0.01$, *** $p < 0.001$, **** $p < 0.0001$.

Supplementary Fig. 1: In vivo delivery of LNPs encapsulating barcoded pDNA. The bar graph illustrates the percentage quantification of LNP (B1–B15) delivery in different target tissues 4 hours after intramuscular injection ($n = 4/\text{group}$). **(A)** Muscle and **(B)** Lymph node **(C)** Liver, **(D)** Heart. Data plotted as mean \pm SEM.

- Can the authors explain how it is controlled that the delivery of each b-DNA-LNP it is not influenced by the presence of the other 14 b-DNA-LNPs?

Response: We thank the reviewer for the thoughtful comment. In our LNP formulation, we incorporated lipid-PEG to enhance colloidal stability and prevent non-specific interactions with other LNPs in vivo. We carefully optimized the molar ratio of lipid-PEG in our formulations based on our previous studies^{1,2}. To further minimize interactions, we pooled the LNPs together just before injection, consistent with our previous studies^{1,2}. Additionally, we measured the hydrodynamic size of the pooled LNPs immediately before injection to ensure homogeneity of the pool."

References:

¹Guimarães, P. P. G. et al. In vivo bone marrow microenvironment siRNA delivery using lipid–polymer nanoparticles for multiple myeloma therapy. *Proceedings of the National Academy of Sciences* 120, (2023).

²Guimaraes, P. P. G. et al. Ionizable lipid nanoparticles encapsulating barcoded mRNA for accelerated in vivo delivery screening. *Journal of Controlled Release* 316, 404–417 (2019).

- And therefore, how is it controlled that promising b-DNA-LNPs are not discarded?

Response: We thank the reviewer for the comment. After evaluating the biodistribution of 15 formulations, we then selected two top-performing LNPs to proceed with in vitro transfection assays with different cell types (Fig. 2A). Based on the delivery results, B2, B3, B4, and B10 LNPs demonstrated enhanced DNA delivery to the spleen, muscle and lymph node (Fig. 1C, D and Supplementary Fig. 1A-B). Because in our approach we are looking for improved specificity, B2 and B3 LNP were not selected due to its concomitant enhanced delivery to the liver and heart, respectively (Supplementary Fig. 1C-D). Beyond specificity, safety concerns were reported recently, related to rare cases of myocarditis and pericarditis after immunization with LNP-based mRNA vaccines from Moderna and Pfizer-BioNTech^{34,35}. Thus, B4 and B10 LNPs were selected as the top-performing LNPs in light of specificity to the spleen and lymph nodes and minimized DNA delivery to the heart and liver compared to other screened LNPs.

References:

³⁴Verma, A. K., Lavine, K. J. & Lin, C.-Y. Myocarditis after Covid-19 mRNA Vaccination. *New England Journal of Medicine* 385, 1332–1334 (2021).

³⁵Husby, A. & Køber, L. COVID-19 mRNA vaccination and myocarditis or pericarditis. *The Lancet* 399, 2168–2169 (2022).

Fig. 1: (C) Heatmap representing delivery to different tissue samples was created using R. Darker clusters indicate higher delivery, whereas lighter clusters indicate lower delivery. Within the heat map, the delivery of different LNP formulations within the same organ (left to right) can be compared. The delivery of the same LNP formulation across different samples (top to bottom) cannot be compared. **(D-E)** The bar graph illustrates the percentage quantification of LNP (B1–B15) delivery in different target tissues 4 hours after intramuscular injection. Data are presented as mean \pm SEM ($n = 4/\text{group}$).

Supplementary Fig. 1: In vivo delivery of LNPs encapsulating barcoded pDNA. The bar graph illustrates the percentage quantification of LNP (B1–B15) delivery in different

target tissues 4 hours after intramuscular injection (n = 4/group). **(A)** Muscle and **(B)** Lymph node **(C)** Liver, **(D)** Heart. Data plotted as mean \pm SEM.

- How homogeneous are the different formulations?

Response: We thank the reviewer for this important point. Each LNP formulation was characterized in three independent experiments, revealing a polydispersity index (PDI) of approximately 0.2 or lower, which indicated the homogeneity of the LNPs. The corresponding PDI values have been added to Table S3. Furthermore, we conducted measurements of the hydrodynamic size of the pooled LNPs immediately before injection, confirming the uniformity of the pool."

Supplementary Table 3: Characterization of b-DNA encapsulated LNPs

LNP formulation	Hydrodynamic diameter (nm)	Polydispersity index (PDI)	Zeta Potential (mV)	
B1	105.7 \pm 1.5	0.114 \pm 0.009	-9.37 \pm 9.15	DOPE
B2	117.1 \pm 4.7	0.187 \pm 0.017	-6.14 \pm 6.02	
B3	111.2 \pm 4.4	0.181 \pm 0.015	-2.2 \pm 6.64	
B4	114.8 \pm 2.9	0.138 \pm 0.030	-1.41 \pm 7.26	
B5	127.2 \pm 1.6	0.141 \pm 0.002	-8.9 \pm 8.75	
B6	107.7 \pm 2.7	0.242 \pm 0.008	-10.05 \pm 7.85	DSPC
B7	96.2 \pm 7.9	0.221 \pm 0.011	-12.35 \pm 6.85	
B8	170.7 \pm 1.0	0.229 \pm 0.009	-9.67 \pm 11.2	
B9	233.3 \pm 6.1	0.224 \pm 0.018	-5.31 \pm 5.41	
B10	183.3 \pm 9.9	0.235 \pm 0.020	1.46 \pm 5.08	DOTAP
B11	117.7 \pm 1.0	0.200 \pm 0.002	-7.84 \pm 6.97	
B12	114.6 \pm 3.2	0.124 \pm 0.011	-23.03 \pm 7.61	
B13	109.8 \pm 1.2	0.102 \pm 0.021	27.53 \pm 4.41	
B14	162.0 \pm 10.7	0.253 \pm 0.016	16.8 \pm 9.55	
B15	84.2 \pm 5.2	0.261 \pm 0.038	20.37 \pm 6.01	

Abbreviations: LNP: ionizable lipid-nanoparticles; nm: nanometer; mV: millivolts; PDI: polydispersity index; DOPE: 1,2-Dioleoyl-sn-glycero-3-phosphoethanolamine; DSPC: 1,2-distearoyl-sn-glycero-3-phosphocholine; DOTAP: 1,2-dioleoyl-3-trimethylammonium-propane.

- Were the different formulations tested individually in-vitro for transfection efficiency, as described for the best two LNPs, B4 and B10?

Response: We thank the reviewer for the comment. Because in our approach we are looking for improved specificity, B4 and B10 LNPs were selected as the top-performing LNPs in light of specificity to the spleen and lymph nodes and minimized DNA delivery to the heart and liver compared to other screened LNPs. Other top-performing LNPs such as B2 and B3 LNP were not selected due to its concomitant enhanced delivery to the liver and heart, respectively (Supplementary Fig. 1C-D). To further refine our

approach, we conducted in vitro studies evaluating the expression induced by our top-performing LNPs in dendritic cells, macrophages, B cells, muscle cells, fibroblasts, and endothelial cells. This assessment indicated the top-performing LNP formulation for subsequent experiments.

Supplementary Fig. 1: In vivo delivery of LNPs encapsulating barcoded pDNA. The bar graph illustrates the percentage quantification of LNP (B1–B15) delivery in different target tissues 4 hours after intramuscular injection (n = 4/group). (A) Muscle and (B) Lymph node (C) Liver, (D) Heart. Data plotted as mean ± SEM.

Fig. 2: Top-performing LNPs induced enhanced in vitro GFP expression in different cell lines. (A) Schematic of the 2 top-performing LNPs encapsulating DNA encoding GFP for in vitro screening of transfection efficiency in four different cell lines. **(B, F, J, N)** Quantification of GFP expression was measured after 24 h, 48 h and 72 h, and cell viability was measured after 72 h in **(D)** macrophages (RAW 264.7), **(H)** dendritic cells (JAWSII), and **(L)** fibroblasts (L929), and **(M)** myoblasts (C2C12) transfected with LNP B4 at different doses ($n = 3/\text{group}$). **(C, G, K, O)** Representative GFP fluorescence (Bottom) and overlaid on brightfield (Top) photomicrographs after treatment of LNP B4

in four different cell lines ($n = 4/\text{group}$). **(E, I, M, Q)** Quantification of GFP expression after treatment with B4 and B10 LNPs in different cell lines. Data are presented as mean \pm SEM ($n = 4/\text{group}$). Data are presented as mean \pm SEM. **(B, F, J, N D, H, L, P)** One-way ANOVA with Dunnet's post hoc test; * $p < 0.05$, *** $p < 0.001$ **** $p < 0.0001$. **(E, I, M, Q)** Two-tailed unpaired t-test; ** $p < 0.01$, *** $p < 0.001$, **** $p < 0.0001$.

Supplementary Fig. 2: Top-performing LNPs induced enhanced *in vitro* GFP expression in endothelial and B cells. **(A)** Schematic of the 2 top-performing LNPs encapsulating DNA encoding GFP for *in vitro* screening of transfection efficiency in endothelial and B cells. **(B)** Quantification of GFP expression was measured after 24 h, 48 h, and 72h. **(C)** Representative GFP fluorescence (Bottom) and overlaid on brightfield (Top) photomicrographs after treatment of LNP B4 in endothelial cells ($n = 4/\text{group}$). **(D)** Cell viability was measured after 72 h in endothelial cells transfected with LNP B4 at different doses ($n = 3/\text{group}$). **(E)** Quantification of GFP expression after treatment with B4 and B10 LNPs in endothelial cells. **(F)** Representative density plots illustrating the gating strategy for CD45+CD3-CD19+GFP+ in splenocytes treated with B4 or B10. **(G)** Quantification of GFP+ expression following treatment with B4 and B10 LNPs in B cells from mouse splenocytes. Data are presented as mean \pm SEM ($n = 4/\text{group}$). Data are presented as mean \pm SEM. **(B, F, D, G)** One-way ANOVA with Dunnet's post hoc test for 72h compared to control; * $p < 0.05$, *** $p < 0.001$ **** $p < 0.0001$. **(E, G)** Two-tailed unpaired t-test; ** $p < 0.01$, *** $p < 0.001$, **** $p < 0.0001$.

4) The antibody responses are shown at 5 days post-infection (Figure 6). The virus provides an additional dose of antigen and therefore boosts immune responses. To evaluate the immune responses solely elicited by the vaccine:

- The authors should include spike-specific IgG levels and SARS-CoV-2 antibody neutralizing titers pre-challenge. This should be possible as according to the methods,

blood samples were collected at day 36 (pre-challenge). Can these data be added to the manuscript?

Response: We thank the reviewer for the thoughtful comment, and we agree. We have incorporated the results of spike-specific IgG levels and SARS-CoV-2 antibody neutralizing titers pre-challenge from immunized mice and hamsters. The findings reveal superior immunogenicity conferred by our LNP-HPS in comparison to control groups, and comparable immunogenicity to the approved COVID-19 vaccine from Biontech/Pfizer (BNT-mRNA) both pre and post-challenge in mice and hamsters.

Fig. 6: LNP-based DNA vaccine induced potent immune response and protection against SARS-CoV-2 infection and its variants (A) Scheme of immunization. K18-ACE-2 mice were immunized with controls and LNP-HPS, and boosted with equivalent doses 3 weeks later. Immunized mice were challenged with lethal 6×10^4 PFU of SARS-CoV-2 variants Gamma lineage (P.1), sera were collected at 24 h before the infection (pre-infection), and 5 days post-infection for all immunized groups and the lungs were collected 5 days post-infection. Levels of total antigen-specific IgG anti-spike serum of mice at (B) pre-infection, and (C) 5 days post-infection ($n = 5$ /group). (D) Levels of total IgA anti-spike in lung homogenate at 5 days post-infection ($n = 5$ /group). Levels of neutralizing antibodies in sera of immunized mice at (E, F) pre-infection, and (G, H) 5 days post-infection against P.1 ($n = 5$ /group). Levels of neutralizing antibodies in sera of immunized mice at (I, J) pre-infection, and (K, L) 5 days post-infection against Omicron ($n = 5$ /group). Data are presented as mean \pm SEM; ** $p < 0.01$. (F, H, J, L) Kruskal-Wallis followed by Dunn's multiple comparisons test.

Fig. 9: LNP-based DNA vaccine induced potent immune response against SARS-CoV-2 infection and its variants in hamsters. (A) Scheme of immunization. Syrian hamsters were immunized with controls and LNP-HPS, and boosted with equivalent doses 3 weeks later. Immunized mice were challenged with 6×10^5 PFU of SARS-CoV-2 variants Gamma lineage (P.1), sera was collected at 24 h before the infection and 5 days post-infection for all immunized groups. Levels of total antigen-specific IgG anti-spike serum of hamsters (B) at pre-infection, and (C) 5 days post-infection ($n = 5$ /group). Levels of neutralizing antibodies in sera of immunized hamsters at (D, E) pre-infection, and (F, G) 5 days post-infection against P.1 ($n = 5$ /group). Levels of neutralizing antibodies in sera of immunized hamsters at (H, I) pre-infection, and (J, K) 5 days post-infection against Omicron ($n = 5$ /group). Data are presented as mean \pm SEM; ** $p < 0.01$. (E,I,G,K) Kruskal-Wallis followed by Dunn's multiple comparisons test.

- To better appreciate the variability among the different animals, can the author provide the data in the graphs per single animal?

Response: We thank the reviewer for the comment. We have provided the data in the graphs per single animal, as shown in the response above for Figures 6 and 9.

- In lines 234-237 it is stated: We next evaluated the levels of neutralizing antibodies in the sera of immunized mice against SARS-CoV-2 variants of Gamma (P.1) and Omicron (B.1.1.529) lineage. We observed that LNP-HPS induced neutralizing antibodies that protected against SARS-CoV-2 variants of Gamma (P.1) and Omicron (B.1.1.529) lineage (Fig. 6D-E). The word protects can be confused with in-vivo protection. Please consider changing it.

Response: We thank the reviewer for bringing this to our attention. We removed the word as suggested.

Results: ...”We observed that LNP-HPS significantly induced neutralizing antibodies against SARS-CoV-2 variants of Gamma (P.1) and Omicron (B.1.1.529) lineage (Fig. 6D-E)”...

5) The expression levels of antiviral genes and cytokines in lungs immunized mice, were measured by quantitative PCR. The authors indicated that there was enhanced expression levels of TNF and IFN- γ in mice immunized with LNP-HPS compared to control groups and that for the other investigated cytokines there were no significant differences between groups.

Response: We thank the reviewer for the comment. We conducted additional experiments, introducing two new groups—naked DNA and BNT-mRNA. Furthermore, we analyzed the expression of cytokines and chemokines in lung tissue. No changes in mRNA levels of IL-10 and TNF were observed between LNP-HPS, PBS, LNP-C, and pHPS. In contrast, the BNT-mRNA group displayed enhanced mRNA levels of IL-10 and TNF compared to all other groups (Supplementary Fig. 6D, H). Increased mRNA levels of CXCL14 were also found for LNP-HPS, BNT-mRNA, pHPS, and LNP-C compared to the PBS group (Supplementary Fig. 6E). No differences were noted in the mRNA levels of IL-1 β , IL-17, IL-6, and IFN- γ between all groups (Supplementary Fig. 6B, C, F, G, and I). Collectively, no correlation was identified between the levels of cytokines and chemokines and the immunogenicity induced by LNP-HPS vaccination.

Supplementary Fig. 6: Levels of cytokines. (A) Scheme of immunization: K18-ACE-2 mice were immunized with controls and LNP-HPS, and boosted with equivalent doses 3 weeks later. Fold change in the gene expression of the indicated cytokines and chemokines (B) IL-1 β , (C) IL-17, (D) IL-10, (E) CXCL14, (F) IL-6, (G) IFN γ , (H) TNF, and (I) CCL2, as determined by RT-qPCR, in lung homogenates of immunized and controls K18-hACE-2 mice ($n = 3-5$ /group). HPRT was used as a reference gene. Data are

presented as mean \pm SEM. One-way ANOVA followed by Tukey's multiple comparison test. * $p < 0.05$.

- Based on Fig. S5 significance is only reported for TNF against mock control. The most relevant comparisons are against PBS and LNP-C. The comparisons showing significant differences should be described in the results.

Response: We thank the reviewer for the comment. In response, we have included statistical significance information for all groups in the figures and the main manuscript, as outlined in Point 5 and Supplementary Figure 5.

- Is statistic missing in the graph for IFN- γ ? In case there is no significant difference for IFN- γ among the different groups, the statement that expression levels are enhanced compared to other groups is not correct and needs to be adjusted.

Response: We thank the reviewer for the comment. In response, we have included statistical significance information for all groups in the figures and the main manuscript, as outlined in Point 5 and Supplementary Figure 5.

- Similarly, also IL-6 gene expression seems higher in mice immunized with LNP-HPS. Is statistic missing in the graph? IL-6 is a pro-inflammatory cytokine associated with severe disease in COVID-19. This observation may deserve more attention as could flag a safety signal. Consider discussing it.

Response: We thank the reviewer for the comment, and we agree. We conducted additional experiments, introducing two new groups—naked DNA and BNT-mRNA. Furthermore, we analyzed the expression of cytokines and chemokines in lung tissue and we have included statistical significance information for all groups in the figures and the main manuscript, as outlined in Point 5 and Supplementary Figure 5. No differences were noted in the mRNA levels of IL-1 β , IL-17, IL-6, and IFN- γ between all groups (Supplementary Fig. 6B, C, F, G, and I).

6) In the discussion, lines 319-321 it is stated: Also, the remarkable resistance against variant P.1 was confirmed by the reduced, if not absent, viral load in the lung of mice immunized with LNP-HPS, 320 as determined via RT-qPCR, plaque assays, and immunofluorescence (Fig. 4E-I).

- This is not fully reflecting the data as by RT-qPCR (Fig 4G) viral load is clearly detected, but significantly lower compared with the LNP-C group. Please correct the statement. In Fig 4G, the comparison with PBS is missing and looks as the one with LNP-C. For consistency with the other panels, please add it for panel G.

Response: We thank the reviewer for bringing this up to our attention. We have corrected the statement as suggested. Also, the comparison with PBS and all other groups in Fig. 4G were included.

Discussion: Also, the remarkable resistance against variant P.1 was confirmed by the reduced viral load in the lung of mice immunized with LNP-HPS, as determined via RT-qPCR, plaque assays, and immunofluorescence (Fig. 4E-I).

Fig. 4G: SARS-CoV-2 genome copies detected in the lung of the immunized mice at 5 days post-infection ($n = 4-5/\text{group}$)

- For immunofluorescence also the mock group is included. Is there a specific reason for it? Why is it not included for the other readouts?

Response: We thank the reviewer for bringing this up to our attention, and we apologize for not include mock group in all graphs. We have now incorporated the mock group into all of our animal-related experiments.

Fig. 4: LNP-based DNA vaccine protects K18-ACE-2 mice against SARS-CoV-2 variants Gamma lineage (P.1) (A) Scheme of immunization: K18-ACE-2 mice were immunized with controls and LNP-HPS and boosted with equivalent doses 3 weeks later. Immunized mice were challenged with lethal 6×10^4 PFU of SARS-CoV-2 variants Gamma lineage (P.1) 15 days after boost. (B, C) The lethality and weight loss were

monitored for 15 days after challenge ($n = 7/\text{group}$). **(D)** Scheme of immunization wherein the lungs were harvested 5 days post-infection for all immunized groups. **(E, F)** The plaque-forming unit measures the viable Vero cells treated with serum of immunized mice 5 days post-infection ($n = 5/\text{group}$). Dashed line denotes limit of detection. **(G)** SARS-CoV-2 genome copies detected in the lung of the immunized mice at 5 days post-infection ($n = 4-5/\text{group}$). **(H)** Analysis of infection via spike protein in lung sections of immunized mice at 5 days post-infection via immunofluorescence from at least 20 fields of 3 different sections ($n = 5/\text{group}$). **(I)** Representative fluorescence images of lungs marked with anti-Spike antibody. Samples were stained with anti-SARS-CoV-2 Spike protein (RBD) antibody (magenta) and DAPI (blue) and acquired using a 20x objective ($n = 5/\text{group}$). Data are presented as mean \pm SEM; * $p < 0.05$, ** $p < 0.01$, *** $p < 0.001$, **** $p < 0.0001$. **(B)** Log-rank (Mantel–Cox) test. **(C)** Two-tailed, unpaired Spearman correlation to test. **(E, H)** One-way ANOVA with Dunnet's *post hoc* test. **(G)** Kruskal–Wallis followed by Dunn's multiple comparisons test.

Fig. 8: Immunogenicity and efficacy of LNP-based DNA vaccine against SARS-CoV-2 variants Gamma lineage (P.1) in hamsters. (A) Scheme of immunization: Syrian hamsters were immunized with controls and LNP-HPS and boosted with equivalent doses 3 weeks later. Immunized hamsters were challenged with 6×10^5 PFU of SARS-CoV-2 variants Gamma lineage (P.1) 15 days after boost. Lungs were

harvested 5 days post-infection for all immunized groups. **(B)** The weight loss was monitored for 5 days after challenge (n = 5/group). **(C)** SARS-CoV-2 genome copies detected in the lung of the immunized hamsters at 5 days post-infection (n = 4-5/group). **(D, E)** The plaque-forming unit measures the viable Vero cells treated with serum of immunized hamsters 5 days post-infection (n = 5/group). Dashed line indicates limit of detection. **(F)** Histopathological analysis at 20 x magnification of the lungs at 5 days post-infection with the P.1 strain (n = 5/group). Histopathological score for **(G)** pleura, **(H)** bronchi, **(I)** alveoli and **(J)** vessels (n = 5/group). The arrows indicate pleura (pink), bronchi (blue), alveoli (red), edema (yellow), vessels (green) and vessels congestion (orange). Data are presented as mean \pm SEM; *p<0.05, **p<0.01, ***p<0.001. **(B)** Two-tailed, unpaired Spearman correlation to test. **(C)** Kruskal–Wallis followed by Dunn’s multiple comparisons test. **(D)** One-way ANOVA with Dunnett’s post hoc test. **(G–J)** Two-way ANOVA followed by Tukey’s multiple comparison test.

7) In the discussion, lines 325-328 it is stated: Intramuscular vaccination with LNP-HPS induced very high titers of antigen-specific IgG in sera as well as enhanced IgA titers in lung homogenate (Fig. 6B-C), suggesting the role of IgA in the early stage of neutralizing response to SARS-CoV-2.

- As indicated in point 4), this statement can only be supported by showing antibody data (spike-specific IgG and SARS-CoV-2 antibody neutralizing titers) pre-challenge. I hope these data can be added.

Response: We thank the reviewer for this important question. To address this point, we have included the results of SARS-CoV-2 antibody neutralizing titers and spike-specific IgG levels pre-challenge. Additionally, we compared these results to those obtained with naked DNA and approved COVID-19 vaccine from Biontech/Pfizer (BNT-mRNA) in both mice and hamster, **as shown in point 4**. Our findings reveal superior immunogenicity conferred by our LNP-HPS in comparison to control groups, and comparable immunogenicity to the approved COVID-19 vaccine from Biontech/Pfizer (BNT-mRNA) both pre and post-challenge in mice and hamsters.

8) In the methods, please include:

- The sex of the K18-hACE2 mice, now only age is indicated.

Response: We thank the reviewer for the comment. We have incorporated the sex in the methods, title and abstract.

Methods: ...“ Female C57BL/6 mice and female Syrian hamsters were purchased from Biotério Central at UFMG. Human Angiotensin Converting Enzyme transgenic mice (K18-hACE2) in the C57BL/6 background, originally from Jackson Laboratories, 8–10 weeks old, were bred at UFMG animal facilities and only female K18-hACE2 mice were utilized as a model of severe COVID-19. Female Syrian Hamsters, 8–10 weeks old, served as a model of mild COVID-19”...

- Where the animal experiments (in C57BL/6 and K18-hACE2 mice) were performed and the committees that approved it.

Response: We thank the reviewer for the comment. We have incorporated the information in the main manuscript as suggested.

Methodology:
Ethics statement

...” This study received approval from the Ethical Committees on the use of animals in research at the Federal University of Minas Gerais (UFMG). The experiments involving mice and hamsters adhered to institutional guidelines for animal ethics and were approved by the Institutional Ethics Committees at UFMG, specifically Commission on Animal Use (CEUA) 177/2020, 245/2021, and 165/2021, for C57BL/6 mice, K18-hACE2 transgenic mice, and Syrian hamsters, respectively.

Mice, hamsters and viruses

...”Female C57BL/6 mice and female Syrian hamsters were purchased from Biotério Central at UFMG. Human Angiotensin Converting Enzyme transgenic mice (K18-hACE2) in the C57BL/6 background, originally from Jackson Laboratories, 8–10 weeks old, were bred at UFMG animal facilities and only female K18-hACE2 mice were utilized as a model of severe COVID-19. Female Syrian Hamsters, 8–10 weeks old, served as a model of mild COVID-19. The experiments were conducted in accordance with the recommendations in the Guide for the Care and Use of Laboratory Animals of the Brazilian National Council of Animal Experimentation (CONCEA). Infections of K18-ACE2 and Syrian hamsters were performed in the Animal Biosafety Level 3 (ABSL-3) facility at the Institute of Biological Sciences from UFMG. All animals were maintained with 12h light/dark cycle with humidity of 50-58% and temperature of 25°C. The SARS-CoV-2 viral strain used in this study belonged to the lineage P.1 (EPI_ISL_13017802) and Omicron (EPI_ISL_7699344) variants. Viral stocks were propagated in Vero CCL81 in a humidified incubator at 37 °C with 5% CO₂ and monitored for cytopathic effects (CPE) daily up to 72 h. Viruses were titrated in Vero CCL81 cells by plaque forming units (PFU) assay, and viral aliquots were kept at –80 °C until further use”...

9) Nowhere is mentioned on which SARS-CoV-2 lineage is based the spike of the vaccine. Can this be indicated in the manuscript?

Response: We thank the reviewer for the comment. We have incorporated the information as suggested.

Results: Hexapro is a recombinant spike protein, derived from the SARS-CoV-2 wild-type, with additional six-proline substitutions (F817P, A892P, A899P, A942P), which exhibited higher expression than two-proline substitutions²⁸. The two-proline substitution in the S2 subunit was used in the mRNA sequences of RNA vaccines approved²⁹.

References:

²⁸Hsieh, C.-L. et al. Structure-based design of prefusion-stabilized SARS-CoV-2 spikes. *Science* (1979) 369, 1501–1505 (2020).

²⁹Chaudhary, N., Weissman, D. & Whitehead, K. A. mRNA vaccines for infectious diseases: principles, delivery and clinical translation. *Nat Rev Drug Discov* 20, 817–838 (2021).

Reviewer #3 (Remarks to the Author):

In "Nanoparticle-based DNA vaccine induced protective effect against SARS-CoV2 2 variants in preclinical model", Prof. Guimaraes and collaborators developed lipid nanoparticles delivering barcoded-DNA (b-DNA) as vaccines for COVID19. The authors used bDNA encoding the proteins of SARS-CoV2 variants, which current vaccine formulations are partially effective. Lipid nanoparticles have demonstrated high performance for RNA and DNA vaccines against COVID-19. In fact, a COVID-19 DNA vaccine based on lipid nanoparticles has been clinically approved, and various formulations are under intense clinical investigation [The Lancet 399, 1313–1321 (2022);

The Lancet 399, 1281–1282 (2022); Nature 597, 161–162 (2021); etc.]. Current study extends the application of lipid nanoparticles to make DNA vaccines against elusive SARS-CoV2 variants. To identify the formulations, the authors used high-throughput screening, which is common in the development of lipid-based vaccines. Thus, while the findings are interesting, I think the work is merely augmenting the existing knowledge rather than introducing any groundbreaking discoveries. The study's methodology is rigorous and adheres to the typical standards in the field. Nevertheless, the conclusions and assertions made are only partially substantiated by the data, and supplementary evidence is required to further support them. Therefore, I would like to note the following points:

Response: We thank the reviewer for their remarkable feedback on our manuscript, as well as for their time and effort in helping us improve it.

1. In the Abstract: "LNP-HPS elicited a robust protective effect against VOCs, as evidenced by improved clinical outcomes and significantly reduced lethality...". The word "clinical" may be a misspelling. I guess the authors are referring to "pre-clinical"

Response: We thank the reviewer for bringing up this typo and we apologize for the confusion. We have reformulated the abstract and corrected this word in discussion.

Abstract: ...”Next, the lead LNP was used to encapsulate HexaPro plasmid (LNP-HPS) to evaluate protective efficacy against SARS-CoV-2 VOC Gamma lineage (P.1) and the immunogenicity against both P.1 and and Omicron (B.1.1.529) VOCs in female mice and hamsters. LNP-HPS elicited a robust protective effect against P.1, correlating with reduced lethality, decreased viral load in the lungs and reduced lung damage. Importantly, LNP-HPS induced potent humoral and T cell responses against P.1, and generated high levels of neutralizing antibodies against P.1 and Omicron. Our findings indicate that the protective efficacy and immunogenicity elicited by LNP-HPS were comparable to those achieved by the approved COVID-19 vaccine from Biontech/Pfizer in both animal models. Together, these findings suggest that LNP-HPS holds great promise as a vaccine candidate against VOCs”...

Discussion: We demonstrated that immunization with LNP-HPS promoted robust protection against P.1, with improved **preclinical** outcome as well as significantly decreased lethality, viral load and consequently reduced lung damage.

2. In the Introduction section there are several points that the authors should reconsider, as follows:

a. In line 93 the authors indicated "In addition, DNA is cost-effective, easy to produce, more stable than mRNA, and does not require frozen transport, which can be relevant for remote areas and developing nations". Actually, the problem of frozen transport has been recently solved in new LNP formulations, which can be transported at higher temperatures (<https://www.nature.com/articles/s41421-022-00517-9>).

Response: We are thankful to the reviewer for the comment. We have reformulated this sentence, as follows:

Introduction: ...“In addition, DNA is cost-effective, easy to produce, and more stable than mRNA, which can be relevant for remote areas and developing nations^{12,14}” ...

b. The authors mentioned in line 96 "Whereas there is one DNA vaccine approved for COVID-19 (ZyCoV-D) and at least 10 in clinical trials (Phase I-III), such as INO-4800, AG0301-COVID19, and GX98 19N, vaccine strategies using DNA-loaded nanoparticles

remain in pre-clinical studies". These numbers, as well as DNA vaccines for other diseases, such as cancer, actually show DNA vaccine technology is already mature. The authors should reconsider this statement.

Response: We are thankful to the reviewer for the comment. We have reconsidered this sentence, as suggested:

Introduction: ...”While ZyCoV-D is the sole DNA vaccine approved for COVID-19^{15,16}, there are at least 10 others currently in various stages of clinical trials (Phase I-III), including candidates like INO-4800, AG0301-COVID19, and GX-19N^{17,18}. This diversity of candidates demonstrates the maturity of DNA vaccine technology“ ...

3. The dynamic light scattering (DLS) technique used in this study does not provide results below 1 nm. Thus, showing DLS results with higher resolution than 1 nm is incorrect. The authors should correct all the diameters.

Response: We thank the reviewer for bringing up this typo and we apologize for the confusion. We have corrected all the diameters.

Supplementary Table 3: Characterization of b-DNA encapsulated LNPs

LNP formulation	Hydrodynamic diameter (nm)	Polydispersity index (PDI)	Zeta Potential (mV)	
B1	105.7 ± 1.5	0.114 ± 0.009	-9.37 ± 9.15	DOPE
B2	117.1 ± 4.7	0.187 ± 0.017	-6.14 ± 6.02	
B3	111.2 ± 4.4	0.181 ± 0.015	-2.2 ± 6.64	
B4	114.8 ± 2.9	0.138 ± 0.030	-1.41 ± 7.26	
B5	127.2 ± 1.6	0.141 ± 0.002	-8.9 ± 8.75	
B6	107.7 ± 2.7	0.242 ± 0.008	-10.05 ± 7.85	DSPC
B7	96.2 ± 7.9	0.221 ± 0.011	-12.35 ± 6.85	
B8	170.7 ± 1.0	0.229 ± 0.009	-9.67 ± 11.2	
B9	233.3 ± 6.1	0.224 ± 0.018	-5.31 ± 5.41	
B10	183.3 ± 9.9	0.235 ± 0.020	1.46 ± 5.08	DOTAP
B11	117.7 ± 1.0	0.200 ± 0.002	-7.84 ± 6.97	
B12	114.6 ± 3.2	0.124 ± 0.011	-23.03 ± 7.61	
B13	109.8 ± 1.2	0.102 ± 0.021	27.53 ± 4.41	
B14	162.0 ± 10.7	0.253 ± 0.016	16.8 ± 9.55	
B15	84.2 ± 5.2	0.261 ± 0.038	20.37 ± 6.01	

Abbreviations: LNP: ionizable lipid-nanoparticles; nm: nanometer; mV: millivolts; PDI: polydispersity index; DOPE: 1,2-Dioleoyl-sn-glycero-3-phosphoethanolamine; DSPC: 1,2-distearoyl-sn-glycero-3-phosphocholine; DOTAP: 1,2-dioleoyl-3-trimethylammonium-propane.

4. The biodistribution studies lack critical tissues, such as muscle, draining lymph nodes, distant lymph nodes, etc. How is the distribution to these tissues?

Response: We thank the reviewer for this important question, and we agree. We have incorporated the biodistribution of 15 LNP formulations encapsulating unique DNA barcodes (b-DNA-LNPs) in draining lymph nodes and muscle tissue as suggested.

Fig. 1: Lipid nanoparticles (LNPs) encapsulating barcoded pDNA (b-pDNA) for accelerated in vivo delivery screening and identification of lead formulations (A) Schematic of LNPs encapsulating barcoded plasmid DNA (pDNA) for accelerated in vivo delivery screening. Left: LNPs were formulated via microfluidic mixing of an aqueous phase of pDNA and an ethanol phase of lipids. Middle: Schematic formation of LNPs encapsulating pDNA. LNPs were formulated via microfluidic mixing, and each LNP formulation was encapsulated with a unique pDNA barcode. Right: 15 LNP formulations (B1–B15) were formulated by varying the identity and molar ratio of phospholipid (DOPE, DOTAP, or DSPC) and molar ratio of cholesterol. A 0.5 μg pDNA dose of each b-DNA LNP was pooled and administered to C57BL/6 mice intramuscularly as a single injection. 4 h post-injection, b-DNA delivery to the heart, liver, spleen, lung, draining lymph nodes, and muscle were quantified. Organs were harvested 4 h post-injection, and pDNA delivery was quantified using both whole-organ for deep sequencing to identify the top-performing LNP. **(B)** Schematic of immunization with LNP-based DNA vaccine and the efficacy in K18-ACE-2 mice against SARS-CoV-2 variants. Top: K18-ACE-2 mice were immunized with either controls or LNP-HPS and received a booster dose after 3 weeks. Bottom: vaccinated mice displayed decreased lethality post-viral challenge and generated potent cellular and humoral responses against SARS-CoV-2 VOCs, indicating

a strong and comprehensive immune response following vaccination. **(C)** Heatmap representing delivery to different tissue samples was created using R. Darker clusters indicate higher delivery, whereas lighter clusters indicate lower delivery. Within the heat map, the delivery of different LNP formulations within the same organ (left to right) can be compared. The delivery of the same LNP formulation across different samples (top to bottom) cannot be compared. **(D-E)** The bar graph illustrates the percentage quantification of LNP (B1–B15) delivery in different target tissues 4 hours after intramuscular injection. Data are presented as mean \pm SEM ($n = 4$ /group).

Supplementary Fig. 1: In vivo delivery of LNPs encapsulating barcoded pDNA. The bar graph illustrates the percentage quantification of LNP (B1–B15) delivery in different target tissues 4 hours after intramuscular injection ($n = 4$ /group). **(A)** Liver, **(B)** Heart **(C)** Muscle and **(D)** Lymph node. Data plotted as mean \pm SEM.

5. In vitro studies evaluated the expression in macrophages, fibroblasts and endothelial cells. The authors should study more relevant cells to the administration route and the immune response, such as dendritic cells, muscle cells, B cells, etc.

Response: We thank the reviewer for the comment. We executed additional in vitro studies to assess the expression in dendritic cells, muscle cells and B cells induced by our top-performers LNPs, as suggested.

Fig. 2: Top-performing LNPs induced enhanced in vitro GFP expression in different cell lines. (A) Schematic of the 2 top-performing LNPs encapsulating DNA encoding GFP for in vitro screening of transfection efficiency in four different cell lines. (B, F, J, N) Quantification of GFP expression was measured after 24 h, 48 h and 72 h, and cell viability was measured after 72 h in (D) macrophages (RAW 264.7), (H) dendritic cells (JAWSII), and (L) fibroblasts (L929), and (M) myoblasts (C2C12) transfected with LNP B4 at different doses ($n = 3/\text{group}$). (C, G, K, O) Representative GFP fluorescence (Bottom) and overlaid on brightfield (Top) photomicrographs after treatment of LNP B4

in four different cell lines ($n = 4/\text{group}$). **(E, I, M, Q)** Quantification of GFP expression after treatment with B4 and B10 LNPs in different cell lines. Data are presented as mean \pm SEM ($n = 4/\text{group}$). Data are presented as mean \pm SEM. **(B, F, J, N D, H, L, P)** One-way ANOVA with Dunnet's post hoc test; * $p < 0.05$, *** $p < 0.001$ **** $p < 0.0001$. **(E, I, M, Q)** Two-tailed unpaired t-test; ** $p < 0.01$, *** $p < 0.001$, **** $p < 0.0001$.

Supplementary Fig. 2: Top-performing LNPs induced enhanced *in vitro* GFP expression in endothelial and B cells. **(A)** Schematic of the 2 top-performing LNPs encapsulating DNA encoding GFP for *in vitro* screening of transfection efficiency in endothelial and B cells. **(B)** Quantification of GFP expression was measured after 24 h, 48 h, and 72h. **(C)** Representative GFP fluorescence (Bottom) and overlaid on brightfield (Top) photomicrographs after treatment of LNP B4 in endothelial cells ($n = 4/\text{group}$). **(D)** cell viability was measured after 72 h in endothelial cells transfected with LNP B4 at different doses ($n = 3/\text{group}$). **(E)** Quantification of GFP expression after treatment with B4 and B10 LNPs in endothelial cells. **(F)** Representative density plots illustrating the gating strategy for CD45+CD3-CD19+GFP+ in splenocytes treated with B4 or B10. **(G)** Quantification of GFP+ expression following treatment with B4 and B10 LNPs in B cells from mouse splenocytes. Data are presented as mean \pm SEM ($n = 4/\text{group}$). Data are presented as mean \pm SEM. **(B, F, D, G)** One-way ANOVA with Dunnet's post hoc test for 72h compared to control; * $p < 0.05$, *** $p < 0.001$ **** $p < 0.0001$. **(E, G)** Two-tailed unpaired t-test; ** $p < 0.01$, *** $p < 0.001$, **** $p < 0.0001$.

6. Besides the cytotoxicity, how is the immune response to the lipids making the vaccine?

Response: We thank the reviewer for the comment. We measured the serum cytokines and chemokines IL-6, IL-10, CCL2, IFN- γ , TNF, and IL-12p70 utilizing flow cytometry. Notably, the LNP-HPS, BNT-mRNA, and pHPS immunized groups exhibited decreased levels of IL-6 and IL-12p70 compared to the PBS, LNP-C groups (Supplementary Fig. 5B, G). However, no differences were observed in the levels of IL-10, CCL2, IFN- γ , and

TNF between groups (Supplementary Fig. 5C-F). Additionally, mRNA expression of cytokines and chemokines in lung tissue was assessed using RT-qPCR (Supplementary Fig. 6). No changes in mRNA levels of IL-10 and TNF were observed between LNP-HPS, PBS, LNP-C, and pHPS. In contrast, the BNT-mRNA group displayed enhanced mRNA levels of IL-10 and TNF compared to all other groups (Supplementary Fig. 6D, H). Increased mRNA levels of CXCL14 were also found for LNP-HPS, BNT-mRNA, pHPS, and LNP-C compared to the PBS group (Supplementary Fig. 6E). No differences were noted in the mRNA levels of IL-1 β , IL-17, IL-6, and IFN- γ between all groups (Supplementary Fig. 6B, C, F, G, and I). Collectively, our data indicate a similar response of LNP-HPS compared to the commercial lipid nanoparticle vaccine BNT-mRNA under the conditions examined in this study.

Supplementary Fig. 5 :(B) Serum cytokines and chemokines were quantified in K18-ACE-2 mice were immunized with controls and LNP-HPS and boosted with equivalent doses 3 weeks later. Immunized mice were challenged with lethal 6×10^4 PFU of SARS-CoV-2 variants Gamma lineage (P.1). Levels of cytokines in serum of mice were measured by at 5 days post-infection (n = 5/group). Data are presented as mean \pm SEM. One-way ANOVA followed by Dunnett's multiple comparison test. **p<0.0001.

Supplementary Fig. 6: Levels of cytokines. (A) Scheme of immunization: K18-ACE-2 mice were immunized with controls and LNP-HPS, and boosted with equivalent doses 3 weeks later. Fold change in the gene expression of the indicated cytokines and chemokines (B) IL-1 β , (C) IL-17, (D) IL-10, (E) CXCL14, (F) IL-6, (G) IFN γ , (H) TNF, and (I) CCL2, as determined by RT-qPCR, in lung homogenates of immunized and controls K18-hACE-2 mice ($n = 3-5$ /group). HPRT was used as a reference gene. Data are

presented as mean \pm SEM. One-way ANOVA followed by Tukey's multiple comparison test. * $p < 0.05$.

7. The safety is not demonstrated. The toxicity of the LNPs against the different tissues (muscle, liver, spleen, heart, etc.) should be confirmed.

Response: We thank the reviewer for this important point. We have evaluated the toxicity of our LNPs against the different tissues of mice and larger animals (hamsters). This assessment includes a comparative analysis with naked DNA and the approved COVID-19 vaccine from Biontech/Pfizer (BNT-mRNA). Importantly, our results indicate the absence of any toxicity associated with the LNPs in the examined conditions. Our findings provide an evidence about the safety of LNPs under the conditions examined in this study.

Supplementary Fig. 7: Histopathological analysis of P.1 SARS-CoV-2 infection in K18-hACE2 mice. (A) Scheme of immunization. K18-ACE-2 mice were immunized with controls and LNP-HPS and boosted with equivalent doses 3 weeks later. Immunized mice were challenged with lethal 6×10^4 PFU of SARS-CoV-2 variants Gamma lineage (P.1). Brain, kidney, heart and liver were harvested at 5 days post-infection for all immunized groups for histopathological analysis. **(B)** Histopathological analysis at 20 x magnification of brain, kidney, heart and liver at 5 days post-infection with the P.1 strain ($n = 5/\text{group}$), reveals no significant histological changes between the groups.

Supplementary Fig. 8: Histopathological analysis of P.1 SARS-CoV-2 infection in hamsters. (A) Scheme of immunization. Syrian hamsters were immunized with controls and LNP-HPS and boosted with equivalent doses 3 weeks later. Immunized hamsters were challenged with 6×10^5 PFU of SARS-CoV-2 variants Gamma lineage (P.1). Brain, kidney, heart and liver were harvested at 5 days post-infection for all immunized groups for histopathological analysis. (B) Histopathological analysis at 20 x magnification of brain, kidney, heart and liver at 5 days post-infection with the P.1 strain ($n = 5$ /group), reveals no significant histological changes between the groups.

8. The post-vaccination serum cytokines levels should be determined. Is there a correlation with the cytokines with the antibody titers and the vaccine protection? By examining these relationships, the significance of the study can be improved.

Response: We thank the reviewer for the insightful question. In response to this comment, we have determined the serum cytokines and chemokines levels for all groups. While we observed a correlation between antibody titers and the vaccine protection, we have not found a clear correlation between the determined cytokines, chemokines and the antibody titers and the vaccine protection. We assessed the SARS-CoV-2 antibody neutralizing titers and specific IgG levels pre-challenge and post-challenge against SARS-CoV-2 VOCs and also compared to naked DNA and approved COVID-19 vaccine from Biontech/Pfizer (BNT-mRNA). The outcomes revealed superior immunogenicity and protection of our LNP-HPS vaccine in comparison to naked DNA, with comparable immunogenicity between LNP-HPS and the approved COVID-19 vaccine from Biontech/Pfizer (BNT-mRNA) in both hamsters and mice. In summary, our findings suggest no discernible correlation between cytokine and chemokine levels and the immunogenicity induced by LNP-HPS or BNT-mRNA vaccination.

Supplementary Fig. 5: (B) Serum cytokines and chemokines were quantified in K18-ACE-2 mice were immunized with controls and LNP-HPS and boosted with equivalent doses 3 weeks later. Immunized mice were challenged with lethal 6×10^4 PFU of SARS-CoV-2 variants Gamma lineage (P.1). Levels of cytokines in serum of mice were measured by at 5 days post-infection (n = 5/group). Data are presented as mean \pm SEM. One-way ANOVA followed by Dunnett's multiple comparison test. **p<0.0001.

9. Could the efficacy of the vaccines be tested against different strains of SARS-CoV2 to determine their suitability? This additional experiment could enhance the significance of the research.

Response: We thank the reviewer for the excellent point. While Omicron currently holds significance as a VOC, this variant induces milder disease in mice and hamsters^{37,38}. In contrast, the Gamma lineage (P.1) has exhibited heightened virulence, pathogenicity, and lethality in K18-hACE2 mice compared to other variants³⁹. Consequently, we opted for the Gamma lineage (P.1) to evaluate the protective efficacy of LNP-HPS vaccination. Additionally, we also examined the neutralizing titers against Omicron using sera from immunized mice both before and after infection.

References:

³⁷Halfmann, P. J. et al. SARS-CoV-2 Omicron virus causes attenuated disease in mice and hamsters. *Nature* 603, 687–692 (2022).
³⁸Suzuki, R. et al. Attenuated fusogenicity and pathogenicity of SARS-CoV-2 Omicron variant. *Nature* 603, 700–705 (2022).
³⁹Stolp, B. et al. SARS-CoV-2 variants of concern display enhanced intrinsic pathogenic properties and expanded organ tropism in mouse models. *Cell Rep* 38, 110387 (2022).

REVIEWERS' COMMENTS

Reviewer #1 (Remarks to the Author):

The authors have done an excellent job at addressing my critique.

Reviewer #2 (Remarks to the Author):

The authors have made a substantial effort to improve the manuscript and incorporate the reviewers' comments.

My concerns have been addressed in the revision and I consider the current version suitable for publication in Nature Communications.

Reviewer #3 (Remarks to the Author):

The authors have addressed my concerns. I only have a minor comment:

In my original comment #3, I indicated that the sizes of the LNP should be corrected, as showing DLS results with higher resolution than 1 nm is incorrect. However, the authors did not make the correction. Here, I note this comment again. And I provide an example of how the data should be presented:

105.7 ± 1.5 --- 106 ± 2

REVIEWER COMMENTS AND AUTHORS' ANSWERS (2nd round)

We thank the reviewers for their remarkable feedback on our manuscript, as well as for their time and effort in helping us improve it. We have revised the manuscript in accordance with the comments received, which we believe have significantly improved our manuscript.

Reviewer #1 (Remarks to the Author):

The authors have done an excellent job at addressing my critique.

Response: We thank the reviewers for their remarkable feedback on our manuscript, as well as for their time and effort in helping us improve it.

Reviewer #2 (Remarks to the Author):

The authors have made a substantial effort to improve the manuscript and incorporate the reviewers' comments.

My concerns have been addressed in the revision and I consider the current version suitable for publication in Nature Communications.

Response: We thank the reviewers for their remarkable feedback on our manuscript, as well as for their time and effort in helping us improve it.

Reviewer #3 (Remarks to the Author):

The authors have addressed my concerns. I only have a minor comment:

In my original comment #3, I indicated that the sizes of the LNP should be corrected, as showing DLS results with higher resolution than 1 nm is incorrect. However, the authors did not make the correction. Here, I note this comment again. And I provide an example of how the data should be presented:

105.7 ± 1.5 --- 106 ± 2

Response: We thank the reviewer for bringing this up to our attention, and we apologize for not making the corrections earlier. We have corrected the DLS results in the main manuscript and Supplementary Table 3.

Supplementary Table 3: Characterization of b-DNA encapsulated LNPs

LNP formulation	Hydrodynamic diameter (nm)	Polydispersity index (PDI)	Zeta Potential (mV)	
B1	106 ± 2	0.114 ± 0.009	-9.37 ± 9.15	DOPE
B2	117 ± 5	0.187 ± 0.017	-6.14 ± 6.02	
B3	111 ± 4	0.181 ± 0.015	-2.2 ± 6.64	
B4	115 ± 3	0.138 ± 0.030	-1.41 ± 7.26	
B5	127 ± 2	0.141 ± 0.002	-8.9 ± 8.75	
B6	108 ± 3	0.242 ± 0.008	-10.05 ± 7.85	DSPC
B7	96 ± 8	0.221 ± 0.011	-12.35 ± 6.85	
B8	171 ± 1	0.229 ± 0.009	-9.67 ± 11.2	
B9	233 ± 6	0.224 ± 0.018	-5.31 ± 5.41	
B10	183 ± 10	0.235 ± 0.020	1.46 ± 5.08	
B11	118 ± 1	0.200 ± 0.002	-7.84 ± 6.97	DOTAP
B12	115 ± 3	0.124 ± 0.011	-23.03 ± 7.61	
B13	110 ± 1	0.102 ± 0.021	27.53 ± 4.41	
B14	162 ± 11	0.253 ± 0.016	16.8 ± 9.55	
B15	84 ± 5	0.261 ± 0.038	20.37 ± 6.01	

Abbreviations: b-DNA: barcoded DNA; LNP: ionizable lipid-nanoparticles; nm: nanometer; mV: millivolts; PDI: polydispersity index; DOPE: 1,3-Dioleoyl-sn-glycero-3-phosphoethanolamine; DSPC: 1,3-distearoyl-sn-glycero-3-phosphocholine; DOTAP: 1,3-dioleoyl-3-trimethylammonium-propane.